# Ethanol Extracts from *Torreya grandis* Seed Have Potential to Reduce Hyperuricemia in Mouse Models by Influencing Purine Metabolism

**DOI:** 10.3390/foods13060840

**Published:** 2024-03-09

**Authors:** Jianghui Yao, Enhe Bai, Yanwen Duan, Yong Huang

**Affiliations:** 1Xiangya International Academy of Translational Medicine, Xiangya School of Medicine, Central South University, Changsha 410013, China; 207501001@csu.edu.cn (J.Y.); baienhe@csu.edu.cn (E.B.); 2Institute of Health and Medicine, Hefei Comprehensive National Science Center, Hefei 230093, China

**Keywords:** *Torreya grandis*, uric acid, ROS, functional food ingredients, gut microbiota

## Abstract

The purpose of this study was to evaluate the efficacy of ethanol extracts from *Torreya grandis* seed (EST) as a functional food in hyperuricemia mice. We investigated EST by analyzing its chemical composition. Using a mouse model of hyperuricemia induced by potassium oxonate (PO), we evaluated the effects of EST on uric acid (UA) production, inflammation-related cytokines, and gut microbiota diversity. The primary constituents of EST consist of various flavonoids and phenolic compounds known for their antioxidant and anti-inflammatory properties in vitro. Notably, our findings demonstrate that EST significantly reduced UA levels in hyperuricemia mice by 71.9%, which is comparable to the effects observed with xanthine treatment. Moreover, EST exhibited an inhibitory effect on xanthine oxidase activity in mouse liver, with an IC_50_ value of 20.90 μg/mL (36%). EST also provided protective effects to the mouse kidneys by modulating oxidative stress and inflammation in damaged tissues, while also enhancing UA excretion. Finally, EST influenced the composition of the intestinal microbiota, increasing the relative abundance of beneficial bacteria such as *Akkermansia muciniphila*, *Corynebacterium parvum*, *Enterorhabdus, Muribaculaceae*, *Marvinbryantia*, and *Blautia*. In summary, our research unveils additional functions of *Torreya grandis* and offers new insights into the future of managing hyperuricemia.

## 1. Introduction

UA is the end product of purine catabolism, and its excessive presence in the body leads to hyperuricemia [1]. It primarily results from both overproduction and inadequate excretion of UA [2]. Over the past few decades, the global prevalence of hyperuricemia has been on the rise, with the highest incidence observed in oceanic countries [3,4,5,6,7]. Notably, Asian countries such as China and the Republic of Korea have experienced a substantial increase in hyperuricemia cases [8,9]. Gout is a common manifestation of hyperuricemia, but there is also an “asymptomatic hyperuricemia” that lacks clinical symptoms like gout or kidney disease [10]. Nevertheless, even in asymptomatic cases, the continuous buildup of UA can lead to the deposition of sodium urate in joints and other tissues, exacerbating health issues. Moreover, hyperuricemia has been linked to various endocrine metabolic disorders, cardiovascular and cerebrovascular conditions, kidney problems, and a range of other diseases, posing significant threats to human well-being [11,12,13,14]. Currently, its treatment primarily aims to alleviate symptoms and enhance the quality of life of patients. Commonly prescribed medications such as colchicine, allopurinol, benzbromarone, and biological agents are utilized for this purpose. However, these treatments often come with limited therapeutic benefits and are associated with adverse reactions like liver injury, nephrotoxicity, bone marrow suppression, and allergic responses [15,16]. Consequently, there is an urgent need for alternative approaches that offer improved efficacy and safety.

Accumulating evidence suggests that changes in intestinal flora often accompany the development of hyperuricemia [17]. For instance, a high-purine diet in mice can alter the intestinal flora, increasing the prevalence of Gram-negative bacteria, which leads to inflammation and hyperuricemia [18]. *Firmicutes* and *Microflora verrucosa* are also commonly dominant bacteria in animals with hyperuricemia and are linked to its development [19]. Hyperuricemia patients exhibit dysbiosis, with elevated aerobic bacteria, *Bacteroides*, and *Escherichia coli*, and reduced levels of *Bifidobacteria* and *Lactobacilli* compared with healthy individuals [20]. Gut microbiota can further influence systemic purine homeostasis and serum UA levels in animal hosts. Bacterial taxa spanning various phyla, including *Bacillota*, *Fusobacteriota*, and *Pseudomonadota*, can utilize purines like UA as sources of anaerobic carbon and energy [21]. Certain gut bacteria, such as *Lactobacillus* and *Pseudomonas*, can break down UA into carbon dioxide and ammonia [22]. *Lactobacillus* can absorb and utilize purine, reducing the absorption of dietary purine in the intestinal tract [23]. Additionally, short-chain fatty acids produced by specific intestinal flora can reduce intestinal inflammation and influence purine metabolism [17,24,25]. For instance, probiotic treatment of hyperuricemia mice using *Clostridium butyricum* not only lowered blood UA levels but also led to the release of intestinal lipopolysaccharides, tumor necrosis factor-α (TNF-α), and inflammatory factors such as interleukin-6. Furthermore, facultative anaerobes produce reactive oxygen species (ROS) using oxygen as the final electron acceptor, promoting xanthine oxidase secretion, thereby increasing UA production [18]. Excessive ROS can also induce the release of inflammatory cytokines, exacerbating hyperuricemia [26,27].

Natural foods and their bioactive compounds have been explored for hyperuricemia treatment, with botanical functional ingredients showing promise in addressing the complex pathogenesis of hyperuricemia [24,28,29]. Phytochemicals like resveratrol, plant saponins, and various plant extracts have demonstrated potential in clinical hyperuricemia treatment [1,30,31,32,33]. These plant ingredients can reduce blood UA levels by modulating enzyme activity, oxidative stress, and proinflammatory and anti-inflammatory cytokines and signaling pathways [34,35]. For example, celery seed extract has been found to inhibit serum UA levels and xanthine oxidase activity in hyperuricemia mice and reduce ankle swelling in rats with acute gouty arthritis induced by intra-articular sodium urate injection [35]. Highly acylated anthocyanins from purple sweet potatoes have also attenuated hyperuricemia and renal inflammation in mouse models [36]. *Melinjo* seed extract reduced serum UA levels in hyperuricemia rats by stimulating the expression of the UA efflux transporter ABCG2 in the intestinal tract [20]. Mangiferin inhibited xanthine oxidase activity and regulated serum superoxide dismutase activity, collectively reducing UA production and oxidative stress [37].

*Torreya grandis* seeds are rich in B vitamins (nicotinic acid and folic acid), mineral elements, and phenolic compounds [38,39]. They are not only enjoyed as delicious fruits but are also used in traditional Chinese medicine to treat conditions such as cough, rheumatism, and intestinal helminthiasis [40]. Furthermore, EST exhibits antioxidant and anti-inflammatory activities [41,42]. There is growing recognition of the gut microbiota’s role in metabolic disorders and emerging interest in natural compounds like EST as potential interventions. Given the nutritional similarities between *Torreya grandis* and the investigated uric acid-lowering plant extracts, as well as their potential impact on intestinal function, we hypothesize that EST may be a functional dietary supplement to control hyperuricemia. Notably, prior to our study, there were no reports regarding the use of *Torreya grandis* seeds for hyperuricemia prevention and treatment. Therefore, the first main aim of this study was to characterize the main components of EST and evaluate its effect in a hyperuricemic mouse model. The secondary goal was to evaluate the EST effect on intestinal microbiota and renal UA excretion. By exploring the interplay between EST, gut microbiota composition, and UA metabolism, we have uncovered novel insights into the functional properties of *Torreya grandis*.

## 2. Materials and Methods

### 2.1. Materials

Torreya grandis seeds were provided from Jiangxi Green Torrey Forestry Co., Ltd. (Nanchang, China) in this study. All standard chemical compounds are above 98% purity and were obtained from Aladdin Biotechnology Co., Ltd. (Shanghai, China). Chromatographic grade acetonitrile and formic acid were purchased from Thermo-Fisher (Waltham, MA, USA). Water used for analysis was filtered through a Milli-Q integral water purification system (Millipore, Danvers, MA, USA). PO (98% potassium oxonate) was sourced from McLean Biochemical Technology Co., Ltd. (Shanghai, China). Allopurinol and CMC-Na (sodium carboxymethyl cellulose 800–1200 mpa. s, USP grade) were obtained from Shanghai Aladdin Biotechnology Co., Ltd. (Shanghai, China). A UA kit was obtained from Wuhan Elabscience Biotechnology Co., Ltd. (Wuhan, China). A xanthine oxidase test kit was obtained from Jingmei Biotechnology Co., Ltd. (Shenzhen, China). Malondialdehyde (MDA), superoxide dismutase (SOD), a glutathione peroxidase (GSH-Px) kit, Interleukin—1β (IL-1β), TNF-α, and a Prostaglandin E2 (PGE2) ELISA kit were purchased from Nanjing Jiancheng Technology Co., Ltd. (Nanjing, China). Analytical pure reagents such as phenylmethanesulfonyl fluoride (PMSF), protein lysate, a bicinchoninic acid (BCA) protein concentration determination kit, Trizol reagent, isopropanol, chloroform, and anhydrous ethanol were purchased from Biyuntian Biotechnology Co., Ltd. (Shanghai, China). Developer and fixer were purchased from Cytiva Life Technology Co., Ltd. (Shanghai, China). Antibodies of anti-rabbit glucose transporter 9 (GLUT9), UA anion transporter 1 (URAT1), organic cation transporter 2 (OCTN2), organic anion transporter 1 (OAT1), and urate secretion transporter 1 (NPT1) were purchased from ABclonal Company (Woburn, MA, USA). The secondary antibody HRP anti-Rabbit IgG was acquired from British Abcam Company (Cambridge, UK) (refer to Table A1 for details).

### 2.2. Methods

#### 2.2.1. Sample Preparation

To investigate the potential nutritional benefits of EST, we conducted an experimental study. Initially, an ethanol extract of dried *Torreya grandis* seeds was prepared. The seeds of *Torreya grandis* were dried in an oven at 60 °C for 24 h. Next, the seeds (1 kg dry weight) were ground into powder in a blender, followed by 3 times sonication 3 times with 75% ethanol (2 L) for 2 h each time. Following sonication, the ethanol extract was concentrated and evaporated under a vacuum. Finally, the resulting extract was lyophilized to 40.1 g of EST for use in further experiments.

#### 2.2.2. Detection and Quantification of Total Flavonoids and Phenols in EST

This study conducted an experimental laboratory-based investigation aimed at determining the chemical composition of EST. The standard curves of total phenols and flavonoids were determined using standard samples from detection kits. The experimental procedure was as follows: Add 2000 μmol/L (300 mg/L) of total phenol standard with 60% ethanol aqueous solution dilute to 1000 μmol/L, 500 μmol/L, 250 μmol/L, and 125 μmol/L. Then, dilute 1 mg/mL of flavonoid standard solution with 60% ethanol to several concentrations of 0.1 mg/mL, 0.08 mg/mL, 0.06 mg/mL, 0.04 mg/mL, and 0.02 mg/mL. The chemical composition was determined through a color reaction involving phenolic substances to reduce tungsten molybdic acid. The compound has a characteristic absorption peak at 760 nm. The absorbance value is measured at 760 nm, and then the total phenolic content of the sample can be obtained. Additionally, the total flavonoid content was measured using a commercial assay, which relies on the radical scavenging activity of flavonoids. In an alkaline nitrite solution, flavonoids react with aluminum ions. This reaction produces a red-colored complex with characteristic absorption peaks at 502 nm. The absorbance at 502 nm was measured to calculate the flavonoid content of the sample.

#### 2.2.3. Liquid Chromatography–Mass Spectrometry (LC-MS) Analysis of EST

This study conducted an analytical investigation utilizing LC-MS analysis to identify and quantify polyphenols present in EST. Comparison with standard reference materials enabled the precise determination of polyphenolic compounds within the extract. Agilent 1100-AB 4000 series instruments (Agilent Ltd., Santa Clara, CA, USA) equipped with Agilent Poroshell 120 EC-C18 2.7 μm (3 × 50 mm) were used for quantitative analyses of the polyphenol constituents of EST. The flow rate was set as 0.6 mL/min and the column temperature was at 35 °C. The mobile phase consisted of formic acid (A) and acetonitrile (C) with the following elution conditions: 0–1 min A-C (95:5); 1–8 min A-C (75:25); 8–12 min A-C (40:60); 12–16 min A-C (0:100); and 16–20 min A-C (95:5). The peak area for EST was acquired using the multiple reaction monitoring mode for quantification.

#### 2.2.4. 1-Diphenyl-2-Picrylhydrazyl (DPPH) Radical-Scavenging Analysis

Following the analytical study, we conducted an experimental investigation to evaluate the free radical scavenging capacity of EST. The DPPH radical scavenging assay was employed, with vitamin C serving as a positive control for comparison. The free radical scavenging activity of EST was determined based on a standard protocol. In brief, EST was first dissolved into methanol and prepared into various concentrations (2.0, 1.0, 0.5, 0.25, 0.125, 0.0625, 0.03125, and 0.01563 mg/mL). The positive controls vitamin C and resorcinol were similarly prepared. Next, the test sample (200 μL) was mixed with DPPH (200 μL, 0.1 mM) and left in a dark environment for 30 min. The absorbance of the resulting solution was measured in a microplate reader at 517 nm. The free radical scavenging activity of the test sample was calculated according to the following formula: clearance capacity (%) = [1 − (A1 − A2)/Ao] × 100%. A1 is the absorbance value of DPPH + sample to be tested; A2 is the absorbance value of methanol + sample to be tested; and A0 is the absorbance value of DPPH + methanol.

#### 2.2.5. Determination of Xanthine Oxidase Activity Inhibition In Vitro

Given that xanthine oxidase is a pivotal enzyme involved in UA production, we conducted an experimental investigation to examine the effect of EST on xanthine oxidase activity. The inhibitory effect of EST on xanthine oxidase inhibition was determined using the xanthine oxidase test kit at 530 nm using an Infinite 200 Pro-Nano Quant multimode microplate reader (Tecan Group Ltd., Männedorf, Switzerland).

#### 2.2.6. Animal Experimental Designs

We next assessed the effects of EST in a mouse model of hyperuricemia. Forty specific pathogen-free (SPF) male Kunming mice (4–6 weeks old, 18–20 g) were obtained and fed at the Experimental Animal Center of Central South University (Changsha, China). The mice were kept in the SPF experimental animal room with a humidity of (50 ± 5) %, a temperature of (21 ± 1) °C, and a day–night interval of 12 h. During the experiment, the mice can freely obtain food and water. A mouse model of hyperuricemia was induced by daily intraperitoneal injections of the uricase inhibitor PO after a week of standard diet adaptation. All mice, except the 7 blank mice in the normal control group (NC group), received intraperitoneal injections of PO daily at a dosage of 300 mg/kg. Two weeks later, blood was taken from the fundus venous plexus of these mice to select the hyperuricemia mice. The twofold increase in UA compared with the normal control mice confirmed the success of the model. Next, the hyperuricemia model mice were randomly divided into three different groups (*n* = 7): the model group (MC group), the positive drug group (AP group), and the EST group. The MC group was composed of the hyperuricemia mice. The purpose of this group was to exclude the possibility that the decrease in UA was due to the action of uric acid oxidase in the mouse body. As a reference, the xanthine oxidase inhibitor, allopurinol, was employed to mitigate UA production. The mice in the AP group received allopurinol treatment at a dosage of 5 mg/kg by oral gavage every day for 2 weeks. Similarly, the mice in the EST group received EST treatment under the same conditions. Mice in the NC and MC groups received 0.5% sodium carboxymethyl cellulose solution by oral gavage. During the treatment period, all hyperuricemia mice continually received PO. At the end of the treatment, all of the mice underwent euthanasia. Their blood, heart, liver, spleen, lung, and kidney were collected and analyzed. Subsequently, we assessed the organ indices of treated mice in each experimental group. The organ index was calculated according to the following formula: organ index (%) = weight of each organ (g)/body weight (g) × 100%.

#### 2.2.7. Biochemical Analysis of Mouse Serum, Liver, and Kidney

To assess kidney function, we examined levels of compounds. Specifically, mouse serum was collected by centrifugation from their blood and used to detect serum creatinine (CREA), blood urea nitrogen (BUN), aspartate aminotransferase (AST), alanine aminotransferase (ALT), cholesterol (CHO), and triacylglycerol (TG) using automatic biochemical detection instruments. Serum UA was detected using a commercial kit. SOD, MDA, and GSH-Px were examined using mouse serum, and the homogenates from the liver, kidney, and intestine of mice were measured using commercial ELISA kits. We conducted an assessment of pain and inflammation-related cytokines, includingIL-1β, TNF-α, and PGE2, in vivo. The IL-1β, TNF-a, and PGE2 in the liver, kidney, and intestine were also examined.

#### 2.2.8. Histopathological Examination

To understand the histological changes in the kidney, we examined the renal tissues of both the control group and the experimental mice. Parts of the heart, liver, spleen, lung, kidney, and intestine of treated mice were fixed in a 4% paraformaldehyde solution, and then dehydrated and embedded in paraffin, which was subsequently stained with hematoxylin and eosin (H&E). Next, the slides were observed under an optical microscope (magnification × 100). The renal histological damage was assessed by semi-quantitative scoring. Blind histopathological examination of kidney tissue was performed by several independent researchers. This method is based on glomerular lesions (from normal glomerular structure to most glomerular atrophy), tubular lesions (from normal tubular structure to most renal tubules), and renal interstitial inflammatory infiltration (from non-inflammatory cells to a large number of inflammatory cells).

#### 2.2.9. Determination of Xanthine Oxidase Inhibition In Vivo

In order to assess enzyme activity levels under varied experimental conditions, mouse liver xanthine oxidase activity was quantified utilizing a commercially available test kit. The activity of the xanthine oxidase enzyme was calculated by detecting the amounts of colored substances. One enzyme activity unit (U) of xanthine oxidase was characterized by catalytic production of 1 nmol colored substance per milligram of protein sample per minute at 37 °C.

#### 2.2.10. Western Blotting Analysis

To delve into the mechanisms by which EST exerts its reno-protective effects in the model mice, we proceeded to investigate the impact of EST on UA excretion in the kidney. Western blotting was used to analyze the amount of β-Actin, URAT1, GLUT9, OCTN2, NPT1, and OAT1 in mouse tissues according to standard procedures. The primary antibodies were diluted in TBST, including antibodies for β-Actin (1:10,000), URAT1 (1:2000), GLUT9 (1:2000), OCTN2 (1:2000), NPT1 (1:2000), and OAT1 (1:2000). The HRP-coupled goat anti-rabbit IgG was used as the secondary antibody (1:10,000) to detect the immune reaction band. The antibodies were visualized by enhanced chemiluminescence after being exposed to X-ray films. The content of the target protein was analyzed with Glyko BandScan 5.0 software (Glyko, Novato, CA, USA) and normalized based on β-Actin.

#### 2.2.11. Gut Microbiota Analysis

To delve deeper into mechanisms of action, we conducted an examination of the potential influence of EST on the gut microbiota of mice. Mouse samples of colonic contents (*n* = 7/group) were first stored in a −80 °C refrigerator. Their total DNA was extracted using a Fast SPIN extraction kit (MP Biomedical, Santa Ana, CA, USA). The V3-V4 region of the bacterial 16S rRNA gene was amplified by PCR. AgencourtAMPure Beads (Beckman Coulter, Indianapolis, IN, USA) were used to purify the resulting amplicons, and a PicoGreensDNA detection kit (Invitrogen, Carlsbad, CA, USA) was used for quantification. Paired-end sequencing was run in an Illumina MiSeq instrument using a MiSeq kit v3 (Shanghai, China). Next, the quality of all reads was scored using quantitative insights into microbial ecology (QIIME2) v1.8.0, and reads with poor quality and short sequences were deleted. The resulting 16 sRNA data were analyzed using QIIME2 (v1.8.0) in the Meiji Microbial Ecological Platform. To assess microbial diversity, we utilized Shannon and Simpson indexes to evaluate α-diversity. For the analysis of microbial population β-diversity, we employed a Principal Coordinates Analysis (PCoA) diagram based on four quadrants. We performed a Principal Component Analysis (PCA) based on un-weighted UniFrac distance. We proceeded to examine the microbial species and their relative abundance at both the phylum and genus levels based on Operational Taxonomic Units (OTUs). The diversity index was calculated by 97% similarity of each OTU. The abundance and diversity of OTUs were studied by PCoA. QIIME2 (v1.8.0) is used to integrate the original sequencing data, OTU classification, α-diversity analysis (including Simpson and Shannon), β-diversity analysis (including PCoA and hierarchical clustering), and microbial composition analysis. To gain further insight into the impact of EST on characteristic bacteria associated with hyperuricemia, we employed linear discriminant analysis effect size (LEfSe) analysis to identify differences in the abundance of four bacterial groups. Furthermore, we utilized PICRUSt 2.0, a bioinformatics software package for predicting metagenomic functions based on marker genes, to investigate differences in metabolic pathways related to changes in fecal microbiota. The Clusters of Orthologous Groups (COGs) function of OTUs was assessed. 

#### 2.2.12. Statistical Analysis

The results are presented as mean ± standard deviation. Analysis of variance was conducted using SAS 9.4 software, followed by the Tukey test for post hoc comparisons. For parametric variables, differences between the two groups were assessed using the unpaired two-tailed Student’s *t*-test. Statistical significance was defined as *p* < 0.05. GraphPad Prism 8 was employed for all statistical analyses.

## 3. Results

### 3.1. Chemical Constituents in EST

The analysis revealed that EST contains an estimated 2290.33 ± 0.77 mg/g of total phenols. Furthermore, the total flavonoid content was determined to be 60.05 ± 3.13 mg/g. The 60 different polyphenols present in EST are presented in Table A2 and Table A3. Notably, the most abundant polyphenol in EST was biochanin A, with a concentration of 27.51 mg/g (Table 1). Other major polyphenols included catechin (6.19 mg/g), epicatechin gallate (6.17 mg/g), and 4-hydroxybenzoic acid (4.51 mg/g). Moreover, EST also contained noteworthy levels of xanthophyll, dihydromyricetin, emodin, carnosic acid, geniposidic acid, caffeine, and 4-methoxycinnamic acid, each exceeding 0.8 mg/g. These findings highlight the diverse and abundant polyphenolic composition of EST, suggesting its potential as a valuable dietary resource.

### 3.2. In Vitro Activity of EST as ROS Scavenger and Xanthine Oxidase Inhibitor

Our results indicated that EST exhibited notable DPPH radical scavenging activity, with an IC_50_ value of 425.7 ± 39.2 μg/mL (Figure 1a). Although this value was lower than that observed for vitamin C (IC_50_ = 10.84 ± 2.54 μg/mL), it nevertheless demonstrated the antioxidant potential of EST. Our findings revealed a dose-dependent inhibition of its activity by EST across concentrations ranging from 10 to 1000 μg/mL (Figure 1b). The dual actions of EST in scavenging free radicals and inhibiting xanthine oxidase activity suggest its potential as a dietary supplement for managing hyperuricemia.

### 3.3. EST Treatment Attenuated Hyperuricemia Symptoms in Model Mice

Figure 1e illustrates that after 2 weeks of PO treatment, the serum UA levels in the model group escalated to approximately 550 μmol/L, marking a 103.2% increase compared with the average UA level of 250 μmol/L in normal mice. This increase confirmed the development of hyperuricemia symptoms in the treated mice. Encouragingly, treatment with EST (5 mg/kg) significantly reduced the serum UA levels in the treated mice to around 150 μmol/L, which is comparable to the levels in mice treated with AP (130 μmol/L).

Hyperuricemia mice displayed a significant increase in BUN levels compared with normal mice, indicating renal dysfunction. Conversely, both EST and allopurinol treatments led to a significant reduction in BUN levels compared with the model group. Furthermore, hyperuricemic mice showed elevated CREA levels compared with normal mice. However, both the EST and AP groups exhibited CREA levels similar to those of normal mice, indicating their protective effects on the kidneys.

Liver function in hyperuricemia mice seemed relatively unaffected, with only slight decreases in serum ALT, CHO, and TG levels and an increase in AST levels (Figure 1f). However, serum and liver levels of antioxidant enzymes, including SOD, MDA, and GSH-Px, were reduced in hyperuricemia mice (*p* < 0.05), while both EST and allopurinol treatments increased these enzyme levels, indicating the potent antioxidant activity of EST and allopurinol. Xanthine oxidase is a key enzyme responsible for UA production. As shown in Figure 1c, the liver xanthine oxidase level significantly increased by 24% in hyperuricemia mice compared with normal mice. Allopurinol, a xanthine oxidase inhibitor, significantly reduced its liver level by 27%, while EST treatment was even more effective, reducing it by 36%, indicating EST’s superior liver protection compared with allopurinol.

The heart and kidney indices were significantly higher in hyperuricemia mice than in the normal group (Table 2). However, EST treatment resulted in a reduction in the heart organ index, suggesting its cardioprotective effects.

### 3.4. EST Improved the Liver and Kidney Enteritis Index IL-1β, TNF-α, PGE2, and Histopathological Changes

The renal tissue of the control group mice displayed normal morphology without any signs of inflammation (Figure 2d). Conversely, the mice treated with PO exhibited histological alterations, including blurred boundaries between adjacent proximal tubular cells, swelling, and proximal tubular necrosis. In comparison with the MC group, the EST treatment group demonstrated improvements in renal tubulointerstitial and glomerular lesions. The cells and cytoplasm in the proximal renal tubules were also notably regulated, mitigating the pathological changes induced by PO.

Our findings indicate that both EST and allopurinol treatments effectively reduced the expression of PGE2 (Figure 2c). Moreover, EST exhibited an additional ability to decrease the expression of IL-1β and TNF-α, underscoring its potential to suppress the inflammatory response and offering protection against PO-induced kidney damage.

### 3.5. The Effects of EST on Protein Levels of Liver, Kidney and Intestine URAT1, GLUT9, OAT1, OCTN2 and NPT1 in Hyperuricemia Mice

In the preceding results, it was observed that EST could effectively reduce the expression levels of inflammatory cytokines PGE2, IL-1β, and TNF-α in PO-induced hyperuricemia mice. To delve into the mechanisms by which EST exerts its reno-protective effects in the model mice, we observed notable changes in renal protein expression (Figure 3b). Specifically, the renal protein levels of OCTN2, OAT1, and NPT1 were significantly reduced, while GLUT9 and URAT1 levels were significantly elevated in mice subjected to PO administration compared with normal mice. Remarkably, both allopurinol and EST treatments resulted in a substantial increase in renal OAT1 protein expression. Furthermore, these interventions effectively upregulated the expression of renal OCTN2 and NPT1 proteins, while downregulating the expression of GLUT9 and URAT1 proteins. This altered protein expression pattern underscores the beneficial impact of both allopurinol and EST in regulating renal UA transporters and highlights their potential to mitigate the effects induced by PO administration.

The liver protein expression levels following allopurinol and EST treatment are illustrated in Figure 3a. After PO administration, the expression level of GLUT9 protein significantly increased. Notably, allopurinol treatment reduced liver GLUT9 protein levels, while EST demonstrated a marked decrease compared with the MC group. Conversely, NPT1 protein expression in the liver was significantly downregulated following PO administration. In contrast, both allopurinol and EST interventions distinctly enhanced liver NPT1 protein expression. This intricate protein expression profile underscores the potential of both allopurinol and EST in modulating liver UA transporters.

The intestinal protein expression levels are presented in Figure 3c. After PO administration, the expression levels of intestinal NPT1 protein significantly decreased. Both allopurinol and EST significantly increased the protein expression of intestinal NPT1. These observations provide crucial mechanistic insights into the action of EST and its potential benefits.

### 3.6. Regulate the Effect of Intestinal Flora in Hyperuricemia Model Mice

#### 3.6.1. Impact on Flora Diversity

The diversity data analysis encompassed 28 samples, resulting in the acquisition of a total of 1,470,365,621,259,835 bases of optimized sequences, with an average sequence length of 423 bp. OTUs were assigned to sequences with an average similarity of ≥97%.

The α-diversity index for the microbiota is presented in Figure 4a,b. Notably, the Shannon index of the EST group and the normal control group was higher than that of the other groups, indicating greater microbial diversity. Conversely, the Simpson index of the EST group and NC group was lower than that of the other groups, suggesting less dominance of specific microbial species. Furthermore, the Venn diagram (Figure 4c) illustrates the overlap of microbial species among the groups. A total of 380 species were identified across all three groups. Additionally, the diagram reveals that 18 species were unique to the NC group, 9 to the MC group, 5 to the allopurinol group, and 10 to the EST group. This highlights the distinct microbial compositions in each group. Finally, the results of PCA (Figure 4d) showed a clear separation between normal mice and hyperuricemia mice, indicating a significant alteration in the microbial community structure of the hyperuricemia mice.

#### 3.6.2. Impact on the Classification and Composition of Intestinal Flora

At the phylum level (Figure 4e), *Bacteroides* emerged as the dominant phylum across all groups. In comparison with the NC group, the proportions of *Actinobacteriota, Campilobacterota*, and *Deferribacterota* in hyperuricemia mice decreased by 74.4%, 29.9%, and 59.2%, respectively. Despite the hyperuricemia condition, *Bacteroides* and *Firmicutes* remained the predominant constituents of the intestinal flora in mice treated with EST. When compared with hyperuricemia mice, *Actinomycetes*, *Desulfobacteriota*, and *Campilobacterota* increased by 5.7-fold, 3-fold, and 4-fold, respectively, in the EST-treated mice. These findings suggest that EST exerts a significant impact on *Actinobacteriota* while having a more limited influence on Firmicutes and Bacteroidota.

Moving to the genus level (Figure 4f), *Lactobacillus*, *Muribaculaceae*, *Lachnospiraceae*_NK4A136_group, and *Lachnospiraceae*_UCG-006 were among the predominant bacteria in the gut of normal mice. The relative ratio of *Muribagulaceae* was significantly higher in hyperuricemia mice compared with normal mice, while the occupancy ratio of *Lachnospiraceae*_NK4A136 was lower in hyperuricemia mice than in normal mice. The microbial community structure of EST-treated mice differed from that of hyperuricemia mice. In the EST group, the relative ratio of *Lachnospiraceae* was also significantly higher than that in both the normal and hyperuricemia mice. A heatmap diagram further illustrates that in the EST group, the abundance of specific genera such as *Corynebacterium parvum*, *Akkermansia muciniphila*, *Enterorhabdus*, *Muribaculaceae*, *Blautia*, *Marvinbryantia*, and *Erysipelato clostridium* increased significantly. These results collectively indicate that EST has beneficial effects on the levels of several genera that are affected by hyperuricemia, further emphasizing its potential in modulating the gut microbiota.

#### 3.6.3. Impact on Key Systemic Types of Intestinal Flora

As depicted in Figure 5a,b, when compared with normal mice, hyperuricemia mice exhibited enrichment in 25 key taxa, with *Roseburia*, *Ruminococcaceae*, *Rikenella*, *Clostridium butyricum*, *Butyricicaceae*, and *Butyricicoccus* being the most prominent. In contrast, *Actinobacteriota*, *Corynebacteria*, *Desulfovibrionaceae*, *Coriobacteria*, *Eggerthellaceae*, and *Enterorhabdus* were the most abundant bacteria in mice treated with EST. Notably, there was a substantial decrease in the relative abundance of *Actinobacteriota* in hyperuricemia mice compared with normal mice (Figure 5c). Importantly, *Actinobacteriota* and *Proteobacteria*, two key taxa with significant distinctions, displayed a significant increase in EST-treated mice (LDA score > 2.0) (Figure 5d). These results indicate that EST treatment can mitigate microbiota changes induced by PO in hyperuricemia mice and modulate specific intestinal microbiota.

In Figure 5e, the COG function of OTUs is displayed. The alterations in the intestinal flora in each group were closely associated with physiological functions, primarily centered around metabolic and genetic information processing pathways. This suggests that EST has the potential to influence gene transcription and translation, initiating a cascade of reactions that ultimately lead to changes in downstream protein conformation and the regulation of UA intake.

## 4. Discussion

We sought to investigate the effects of EST on hyperuricemia. Before our study, the impact of *Torreya grandis* on hyperuricemia had not been explored. As far as our knowledge extends, our research represents the first demonstration that EST can significantly reduce UA levels in the mouse model. Furthermore, we identified and quantitatively analyzed various classes of compounds present in EST, providing valuable insights for future research in the development of disease-specific functional foods. Several studies have highlighted the significance of xanthine oxidase in the treatment of gout [30,43]. Compounds derived from plants, such as anthocyanins, curcumin, flavonoids, and ellagic acid, have been shown to inhibit liver xanthine oxidase activity and reduce blood UA levels in hyperuricemia mice [44,45,46]. The major flavonoids found in *Torreya grandis*, including 4-hydroxy benzaldehyde, 4-methoxypyrocatechol, coniferylaldehyde, 4-hydroxycinnamaldehyde, β-sitosterol, and daucosterol, are known for their antioxidant properties and ability to inhibit enzyme activity and scavenge free radicals [28]. Consequently, it is plausible that EST’s inhibitory effect on xanthine oxidase activity is attributed to these flavonoids. To further elucidate the mechanisms involved, we evaluated EST’s impact on xanthine oxidase activity, and our results indicate that EST can effectively reduce serum xanthine oxidase activity both in vitro and in vivo, shedding light on the mechanisms of EST’s therapeutic action in hyperuricemia.

To establish the hyperuricemia model in rodents, PO is often used due to its ability to effectively inhibit uricase. Consistent with previous studies, we observed that the administration of 300 mg/kg of PO for 14 days led to a significant increase in serum UA levels in model mice, representing a 103.2% elevation compared with normal control mice. Importantly, EST treatment effectively reduced serum UA levels in the mouse model. In some clinical trials, people have observed that the inflammatory state and certain structural changes were prominent pathological features of hyperuricemia [47,48]. To understand the histological changes in the kidney, we examined the renal tissues of both the control group and the experimental mice. Moving beyond the histopathological observations of various organs, we conducted an assessment of pain and inflammation-related cytokines, including IL-1β, TNF-α, and PGE2 in vivo. Significantly, the mice with hyperuricemia display compromised antioxidant systems, resulting in oxidative stress characterized by elevated levels of MDA and decreased activity of SOD and GPX in both blood and liver. However, treatment with EST demonstrated a notable increase in SOD and GPX activity and a reduction in MDA content compared with the untreated group, indicating EST’s potential to mitigate oxidative stress in hyperuricemic mice.

Further investigation was carried out to analyze the urate excretion pathway in mice exhibiting hyperuricemia. The regulation of renal UA involves various transporters, including URAT1, GLUT9, OAT1, NPT1, and OCTN2 [49,50]. URAT1 and GLUT9 reabsorb and transport UA, while OAT1 facilitates its secretion into tubular cells, NPT1 promotes excretion into urine, and OCTN2 plays a role in renal function [51,52,53]. Our findings revealed that EST can upregulate the protein expression of OCTN2 in the kidney of hyperuricemia mice, suggesting its potential for improving renal function. Therefore, this organic transporter may be a significant target of EST in hyperuricemia treatment. Since more than 90% of excreted UA is reabsorbed and only about 10% is excreted in urine, regulation of UA reabsorption plays a pivotal role in UA excretion. URAT1 and GLUT9 are considered attractive therapeutic targets for hyperuricemia [51]. Our results indicate that EST can reduce UA uptake via URAT1 and GLUT9 transporters while reducing the expression of renal transporter OAT1, thereby inhibiting UA reabsorption and promoting UA excretion, ultimately alleviating hyperuricemia. Therefore, EST holds great potential for facilitating UA excretion.

The above results confirm the significant role of EST in ameliorating hyperuricemia. To explore how EST regulates intestinal function and promotes UA excretion, we conducted an examination of the potential influence of EST on the gut microbiota of mice. Compared with normal mice, hyperuricemia mice exhibited lower levels of *Actinobacteria*, *Campylobacter*, and *Desulfobacteria* in their intestinal flora. Interestingly, the administration of EST significantly increased the levels of *Actinobacteria*, *Campylobacter*, and *Desulfobacteria*, as well as *Trichospirillum* and *Erysipelothrix*. These bacteria are involved in carbohydrate metabolism, producing short-chain fatty acids with immunomodulatory and anti-inflammatory effects on the mouse intestine [17,54]. Additionally, EST treatment led to a significant increase in *Akkermansia muciniphila*, *Corynebacterium parvum*, *Enterorhabdus*, *Muribaculaceae*, *Marvinbryantia*, and *Blautia*. *Akkermansia muciniphila*, often referred to as the “next-generation probiotic”, enhances cancer immunotherapy responses and vitamin K_2_ levels [55]. *Corynebacterium parvum* also has immune-enhancing and anti-tumor properties [56]. *Enterorhabdus*, *Muribaculaceae*, *Marvinbryantia*, and *Blautia* contribute to colon protection and have a positive association with tight junction proteins, negatively impacting inflammatory factors [57]. These microorganisms hold promise as potential probiotics capable of preventing inflammation, promoting short-chain fatty acid production, and maintaining intestinal stability [58]. Collectively, EST can mitigate structural changes in the intestinal flora of hyperuricemia mice. This modulation of gut microbiota not only underscores the potential of EST in UA metabolism but also highlights its broader impact on gut health.

The limitations of our study include the following: First, only a single concentration of EST, i.e., 5 mg/kg, was used in the mouse study, and thus a dose-dependent effect of EST against hyperuricemia would further strengthen the conclusion of our study; second, it is important to acknowledge that EST likely comprises many additional components and compounds beyond flavonoids and polyphenols and further research is imperative to analyze and elucidate the mechanism of these bioactive compounds.

This study elucidates the potential of polyphenol-rich foods in promoting health. The composition of *Torreya grandis* unveils novel therapeutic avenues for high UA treatment and uncovers additional medicinal functions. Moreover, exploring the underlying mechanisms of EST’s effects could unveil new insights into UA regulation and microbiota modulation. It offers new insights into preventing and managing various chronic diseases, especially for targeted interventions and personalized treatment approaches. The reduction in serum UA levels within the EST group compared with the model group is approximately 72.73%, while within the AP group, it approximates 76.36%. Moreover, allopurinol achieved a notable reduction in liver xanthine oxidase levels by 27%. Although not as potent as allopurinol in conventional UA therapy, EST treatment resulted in a reduction in liver xanthine oxidase levels by 36%, implying superior hepatic protection compared with allopurinol. Furthermore, EST presents advantages of cost-effectiveness and cultural acceptance, particularly in resource-constrained regions or where traditional herbal medicine is prevalent. It holds promise as a safer, more affordable, and sustainable alternative for hyperuricemia treatment. Despite demonstrating anti-gout effects linked to antioxidants and anti-inflammatory properties, the specific components in EST may vary. Further systematic exploration is warranted to identify the extract’s active therapeutic agent. Clinical validation remains imperative to establish the efficacy of EST in preventing or treating hyperuricemia.

## 5. Conclusions

In this study, EST exhibits a multifaceted impact on UA regulation. It effectively reduces UA production by inhibiting xanthine oxidase activity in both the blood and the liver. Additionally, it modulates key urate transporters, including GLUT9, URAT1, OCTN2, OAT1, and NPT1, to curb UA reabsorption and promote UA excretion. Furthermore, EST administration significantly improves the diversity of gut microbiota, with notable increases in the proportions of *Akkermansia* and *Corynebacterium parvum*. This modulation not only affects urate metabolism but also enhances gut health. Collectively, our study provides valuable insights into the multifaceted mechanisms by which EST can mitigate hyperuricemia, making it a promising candidate for further research and potential therapeutic applications.

## Figures and Tables

**Figure 1 foods-13-00840-f001:**
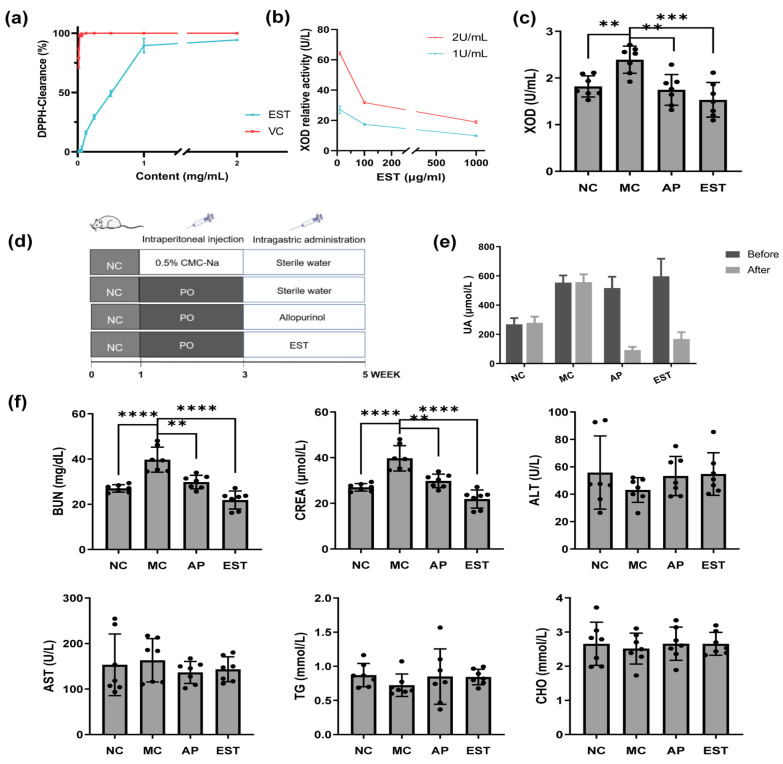
Effects of EST in vivo and in vitro. Antioxidant activities of the ethanol extract from *Torreya grandis* seeds. (**a**) DPPH radical scavenging activity; (**b**) xanthine oxidase activity inhibition ability of EST in vitro; (**c**) xanthine oxidase activity inhibition ability of EST in liver; (**d**) graphical presentation of the experimental design; (**e**) uric acid level of the ethanol extract from the seeds of *Torreya grandis* in PO-induced hyperuricemia mice, with the results presented as the mean ± SEM of eight mice; and (**f**) EST and allopurinol impact on BUN, CREA, ALT, AST, CHO, and TG levels. The * marked by different letters represents significantly different results at the level of ** *p* < 0.01, and *** *p* < 0.001, **** *p* < 0.0001, which represents the MC group vs. other groups according to one-way ANOVA followed by the Bonferroni multiple comparison test.

**Figure 2 foods-13-00840-f002:**
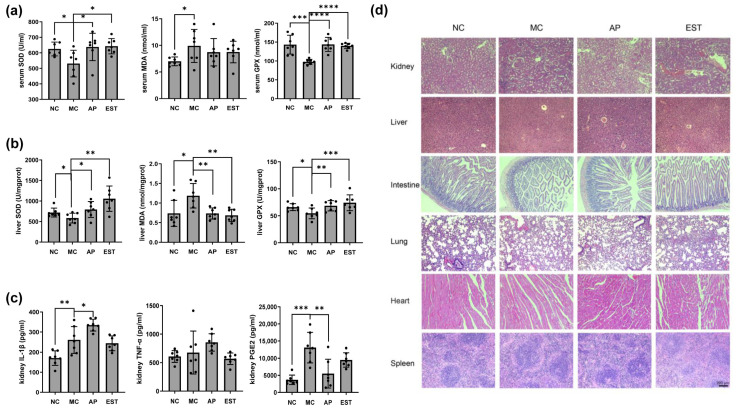
Effects of EST on the physiological status of mice with hyperuricemia. (**a**) EST and allopurinol effects on SOD, MDA, and GSH-Px levels in serum; (**b**) EST and allopurinol effects on SOD, MDA, and GSH-Px levels in liver; (**c**) EST and allopurinol effects on kidney IL-1β, TNF-α, and PGE2 level; and (**d**) hematoxylin and eosin-stained section of mouse organ compared with four groups (using optical microscope (Leica Microsystems CMS GmbH, Wetzlar, Germany) original magnification ×100). The * marked by different letters represents significantly different results at the level of * *p* < 0.05, ** *p* < 0.01, and *** *p* < 0.001, **** *p* < 0.0001, which represents the MC group vs. the other groups according to the one-way ANOVA followed by the Bonferroni multiple comparison test.

**Figure 3 foods-13-00840-f003:**
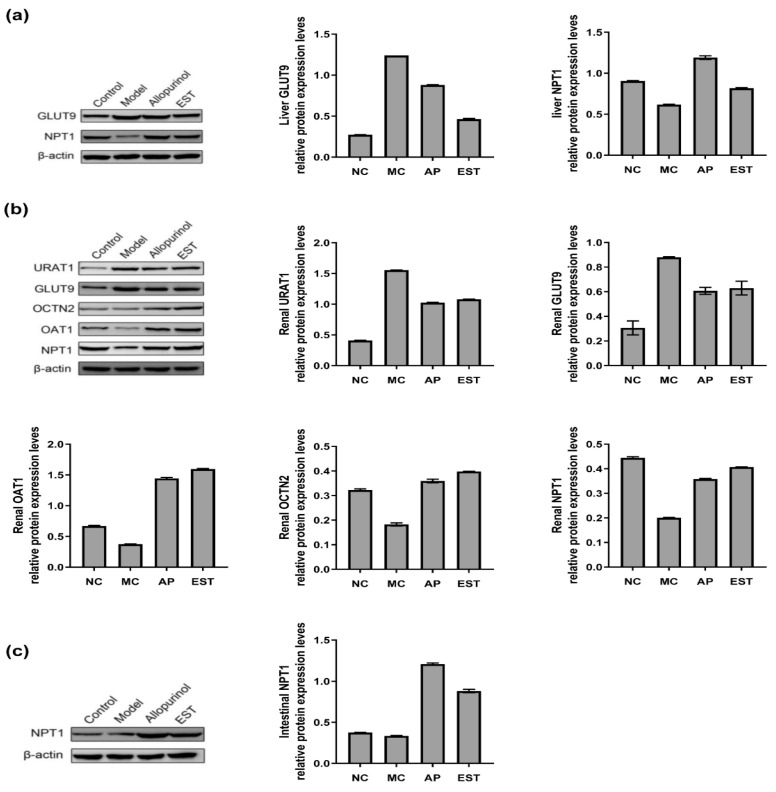
EST affects transport protein expression in PO-induced hyperuricemia. (**a**) EST treatment effects on protein expression of mouse GLUT9 and NPT1 in isolated liver tissues; (**b**) EST treatment effects on protein expression of mouse URAT1, GLUT9, OCTN2, OAT1, and NPT1 in isolated renal tissues; and (**c**) EST treatment effects on protein expression of mouse NPT1 in isolated intestine tissues. Data are presented as mean ± SEM (*n* = 3).

**Figure 4 foods-13-00840-f004:**
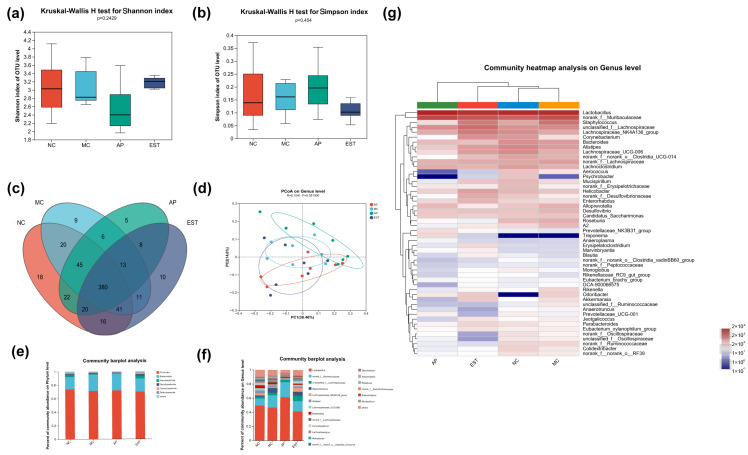
Analysis of intestinal flora composition in mice. (**a**) Shannon index of four groups; (**b**) Simpson index of four groups; (**c**) Venn diagram of four groups; (**d**) PCA plot based on genus level; (**e**) relative community abundance of flora at the phylum level; (**f**) relative community abundance of flora at the genus level; and (**g**) community heatmap at the genus level.

**Figure 5 foods-13-00840-f005:**
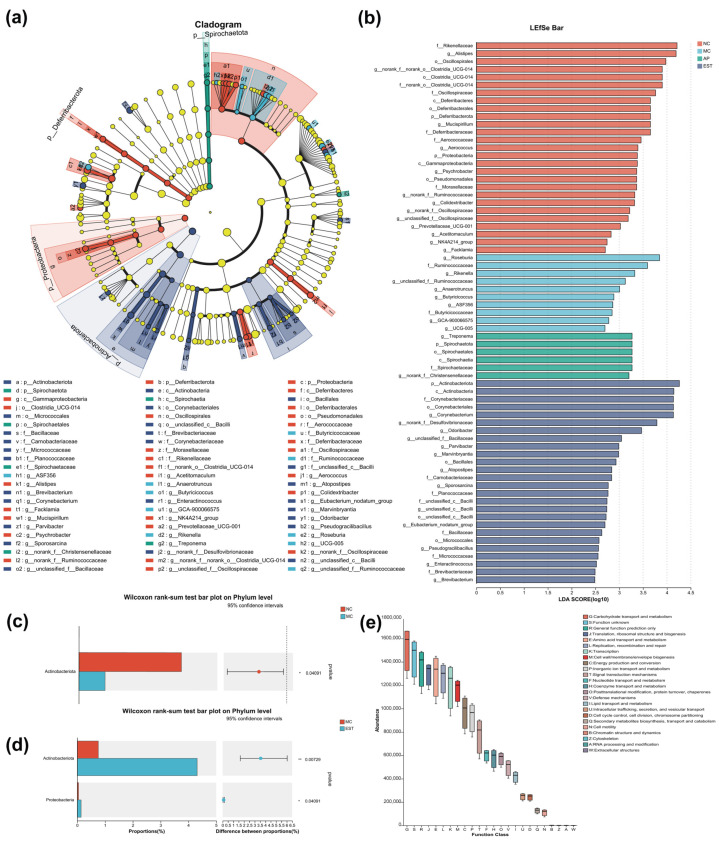
Analysis of the variability in intestinal flora in mice. (**a**) Evolutionary branching diagram for LEfSe analysis; (**b**) histogram of LDA distribution for LEfSe analysis; (**c**) comparison of bacterial microbiota between NC and MC at the genus level; and (**d**) comparison of bacterial microbiota between MC and EST at the genus level (* *p* < 0.05, ** *p* < 0.01). (**e**) The COG function of OTUs was determined as follows: G: transport and metabolism of carbohydrates; E: transport and metabolism of amino acids; P: transport and metabolism of inorganic ions; J: translation, ribosome structure and biogenesis; L: replication, recombination and repair; K: transcription; M: biogenesis of cell wall and membrane; C: energy production and conversion; and T: signal transduction mechanism.

**Table 1 foods-13-00840-t001:** Polyphenol compounds of EST.

Polyphenols	Calculated Concentration (mg/L)
Biochanin A	68.78 ± 59.27
Catechin	15.47 ± 12.52
Epicatechin gallate	15.42 ± 12.44
4-hydroxybenzoic acid	11.28 ± 15.72
Xanthophyll	8.75 ± 4.07
Dihydromyricetin	6.72 ± 1.74
Emodin	4.13 ± 3.07
Carnosic acid	3.86 ± 3.22
Geniposidic acid	3.43 ± 2.09
Caffeine	2.29 ± 2.35
4-methoxycinnamic acid	2.16 ± 1.42
Trans-cinnamic acid	1.97 ± 1.54
Procyanidin b1	1.92 ± 1.59
β-Sitosterol	1.48 ± 0.76
Sinapic acid	1.31 ± 0.47
Salvianic acid A	1.03 ± 0.06
Phenylalanine	1.00 ± 0.03
Resveratrol	0.99 ± 0.04
Sinapine	0.98 ± 0.01
Gallocatechin	0.95 ± 0.01

Enumeration of the predominant top 20 polyphenolic compounds by content. Each value was recorded as the mean value ± standard deviation of two replicates.

**Table 2 foods-13-00840-t002:** Body mass index of each organ.

Group	Heart (%)	Liver (%)	Spleen (%)	Lung (%)	Kidney (%)
NC	0.52 ± 0.06 *	4.40 ± 0.39	0.36 ± 0.07	0.61 ± 0.06	1.30 ± 0.09 *
MC	0.61 ± 0.04	4.39 ± 0.44	0.42 ± 0.23	0.59 ± 0.09	1.45 ± 0.13
AP	0.54 ± 0.04 *	4.39 ± 0.56	0.39 ± 0.07	0.53 ± 0.04	1.37 ± 0.18
EST	0.57 ± 0.08 **	4.18 ± 0.10	0.35 ± 0.18	0.59 ± 0.06	1.40 ± 0.10

Each value was recorded as the mean value ± standard deviation of seven replicates. * A superscript asterisk on the value indicates a significant difference between the MC group and the other groups (* *p* < 0.05. ** *p* < 0.01).

## Data Availability

The original contributions presented in the study are included in the article, further inquiries can be directed to the corresponding authors.

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
