# Peer review of "Ethanol Extracts from Torreya grandis Seed Have Potential to Reduce Hyperuricemia in Mouse Models by Influencing Purine Metabolism"

_foods, 2024, doi:10.3390/foods13060840_

Round 1

Reviewer 1 Report

Comments and Suggestions for Authors

though this experimental study sounds really interesting, there are some important points that reduce the quality of the information provided. Some points need clarification, refinement, rewriting and more information to improve this article. Several changes need to be made.

1.      The title should be improved and more specific, such as “Ethanol extracts from Torreya grandis seed have potential to reduce hyperuricemia in mouse models”. In the abstract: What does EST mean? edible seeds of T. grandis? Only the first time that an abbreviation appears, the full name should be entered. It seems to me that the M&M section has been written after the results and discussion section, and this is the reason why most of the abbreviations now appear after and not before, as this magazine recommends. Please check the proper placement of the full names of all abbreviations. Before the results, it would be better to write the main aim of this experimental study and summarize the methodology. The main goal must be the same throughout the manuscript. Authors should not use the words that appear in the title as keywords, e.g., Torreya grandis, Hyperuricemia. References must be recent and relevant.

2.      The introduction section needs to be improved. The research question should be clearly outlined. A good and clear justification for conducting this study should be given. It is not clear what the authors tried to convey. The authors need to relate the topics covered (Hyperuricemia/Gout, microbiota, EST) to establish a hypothesis and the main objective of this experimental study. There is too much information that is not substantial to establish a hypothesis/objective. Please delete “The systemic…flora”. It would be a good idea to summarized in one paragraph the lines 52-78. Why is this study important? Hypothesis, for example, “We hypothesize that EST may be a functional dietary supplement to control hyperuricemia. Therefore, the first main aim of this study was to characterize the main components of EST and evaluate its effect in hyperuricemic mouse model. The secondary goal was to evaluate the EST effect on intestinal microbiota and renal uric acid excretion. This section should be summarized in three to five paragraphs.

3.      The materials and methods section needs improvements. What type of study was it? Please check whether the full name of the abbreviations has been written above or not. What does AP group mean? If there were 40 SPF males and there are four groups with 10 mice in each, what were the remaining mice? Did the AP group receive allopurinol and the EST group receive EST as treatment? What does CMC mean? There is key information that appears in the results section (which has been crossed out), that would improve the description of the design of this study and that must be added in the M&M section in the appropriate subsections: To explore the potential nutritional benefits of EST, we initially prepared an ethanol extract of dried T. grandis seeds. Its chemical composition was determined through a colorimetric reaction involving phenols and ferric ions. The total flavonoid content was measured using a commercial DPPH assay, which relies on the radical scavenging activity of flavonoids. Through high-resolution liquid chromatography-mass spectrometry (HR-LCMS) analysis and comparison with standard reference materials, we identified and quantified polyphenols present in EST. We next assessed the capacity of EST to scavenge free radicals, employing the DPPH radical scavenging assay and using vitamin C as a positive control. Given that XOD is a pivotal enzyme involved in UA production, we further examined the effect of EST on XOD activity. We next assessed the effects of EST in a mouse model of hyperuricemia, which was induced by daily intraperitoneal injections of the uricase inhibitor potassium oxonate (PO). As a reference, the XOD inhibitor, allopurinol (AP), was employed to mitigate UA production. To assess kidney function, we examined creatinine (CREA) and blood urea nitrogen (BUN) levels. Subsequently, we assessed the organ indices of treated mice in each experimental group. To understand the histological changes in the kidney, we examined the renal tissues of both the control group and the experimental mice. We conducted an assessment of pain and inflammation-related cytokines, including interleukin - 1β (IL-1β), tumor necrosis factor-α (TNF-α), and prostaglandin E2 (PGE2), in vivo. To delve into the mechanisms by which EST exerts its reno-protective effects in the model mice, we proceeded to investigate the impact of EST on UA excretion in the kidney. To delve deeper into mechanisms of action, we conducted an examination of the potential influence of EST on the gut microbiota of mice. To assess microbial diversity, we utilized Shannon and Simpson indexes to evaluate α-diversity. For the analysis of microbial population β-diversity, we employed a Principal Coordinates Analysis (PCoA) diagram based on four quadrants. We perform a Principal Component Analysis (PCA) based on un- weighted UniFrac distance. We proceeded to examine the microbial species and their relative abundance at both the phylum and genus levels based on Operational Taxonomic Units (OTU). To gain further insight into the impact of EST on characteristic bacteria associated with hyperuricemia, we employed linear discriminant analysis effect size (LEfSe) analysis to identify differences in the abundance of four bacterial groups. Furthermore, we utilized PICRUSt, a bioinformatics software package for predicting metagenomic functions based on marker genes, to investigate differences in metabolic pathways related to changes in fecal microbiota. The Clusters of Orthologous Groups (COG) function of OTUs was assessed. The description in Statistical Analysis should be improved.

4.      In the results section: In this section and according to the study design specified above, only the results should be described. In the text, the authors should write the most significant results, and they should avoid repeating the same information in the text if this data appear in the tables or figures. The quality of figures should be improved to see the results adequately.

5.      The discussion needs improvements. This section should start with the main objective of this study and the most significant results. It would be better if the authors analyzed/discussed their results based on the steps/phases of the study design/results. Why this study is important/crucial/essential to understand the mechanism of a possible treatment for EST. Were these compounds found in the EST of this study? Compared to the usual treatment for hyperuricemia (allopurinol, etc.), what advantages/disadvantages would EST offer as a treatment/supplement for hyperuricemic patients? It would be good to write a paragraph about suggestions of the knowledge reached so far before the conclusions. What would be the next step?

6.      Conclusion should be improved. Delete what is crossed out here.

I would like to encourage the authors to rewrite this manuscript following the natural flow of knowledge on this important topic.

Comments on the Quality of English Language

Minor editing of English language required.

Author Response

Response to Reviewer 1 Comments

1. Summary

2. Point-by-point response to Comments and Suggestions for Authors

Comments 1: The title should be improved and more specific, such as “Ethanol extracts from Torreya grandis seed have potential to reduce hyperuricemia in mouse models”. In the abstract: What does EST mean? edible seeds of T. grandis? Only the first time that an abbreviation appears, the full name should be entered. It seems to me that the M&M section has been written after the results and discussion section, and this is the reason why most of the abbreviations now appear after and not before, as this magazine recommends. Please check the proper placement of the full names of all abbreviations. Before the results, it would be better to write the main aim of this experimental study and summarize the methodology. The main goal must be the same throughout the manuscript. Authors should not use the words that appear in the title as keywords, e.g., Torreya grandis, Hyperuricemia. References must be recent and relevant.

Response 1: Thank you for your valuable feedback. Here's how we address your suggestions:

(1)    Title Improvement: Thank you for the suggestion. We have revised the title to make it more specific and informative, as follows: " Ethanol extracts from Torreya grandis seed have potential to reduce hyperuricemia in mouse models by influencing purine metabolism ".

(2)    EST Abbreviation in Abstract: We apologize for the oversight. EST stands for "Ethanol Extracts from Torreya grandis Seeds," and we have ensured that the full name is provided the first time the abbreviation is used in the abstract. (Lines 12,15)

(3)    Methods and Materials Section. We appreciate your observation. We have rearranged the Methods and Materials section to adhere to the recommended format, ensuring that abbreviations appear before their use in the Results and Discussion sections.

(4)    Consistency in Abbreviations. We have meticulously checked and ensured that the full names of all abbreviations are provided before their first use in the manuscript.

(5)    Clarity of Main Aim and Methodology. We have added the main aim of the study and provided a succinct summary of the methodology before presenting the results, ensuring coherence and clarity throughout the manuscript: “The purpose of this study was to evaluate Ethanol extracts from Torreya grandis seed (EST) efficacy as a functional food in hyperuricemia mice. We investigated the EST by analyzing its chemical composition. Using a mouse model of hyperuricemia induced by potassium oxonate (PO), we evaluated effects of EST on uric acid (UA) production, inflammation-related cytokines and gut microbiota diversity.” (Lines 12-16)

(6)    Keyword. We have revised the keywords to exclude terms already present in the title as per your recommendation: Merrillii, Uric acid, ROS, Phytochemicals, gut microbiota. (Lines 23)

(7)    Recent and Relevant References. We appreciate the attention to the references cited in our manuscript. Ensuring that our paper includes the most recent and relevant studies is crucial for providing readers with up-to-date information. We have reviewed and updated the references to include recent and pertinent literature relevant to the study topic: (Lines 595, 598, 601,614,660,663)

3Lin, Z., Chen, H., Lan, Q., Chen, Y., Liao, W., & Guo, X. (2023). Composite Dietary Antioxidant Index Is Negatively Associated with Hyperuricemia in US Adults: An Analysis of NHANES 2007-2018. International journal of endocrinology, 2023, 6680229. https://doi.org/10.1155/2023/6680229

4Almuqrin, A., Alshuweishi, Y. A., Alfaifi, M., Daghistani, H., Al-Sheikh, Y. A., & Alfhili, M. A. (2024). Prevalence and association of hyperuricemia with liver function in Saudi Arabia: a large cross-sectional study. Annals of Saudi medicine, 44(1), 1825. https://doi.org/10.5144/0256-4947.2024.18.

5Otaki, Y., Konta, T., Ichikawa, K., Fujimoto, S., Iseki, K., Moriyama, T., Yamagata, K., Tsuruya, K., Narita, I., Kondo, M., Shibagaki, Y., Kasahara, M., Asahi, K., & Watanabe, T. (2021). Possible burden of hyperuricaemia on mortality in a community-based population: a large-scale cohort study. Scientific reports, 11(1), 8999. https://doi.org/10.1038/s41598-021-88631-8

11Vareldzis, R., Perez, A., & Reisin, E. (2024). Hyperuricemia: An Intriguing Connection to Metabolic Syndrome, Diabetes, Kidney Disease, and Hypertension. Current hypertension reports, 10.1007/s11906-024-01295-3. Advance online publication. https://doi.org/10.1007/s11906-024-01295-3

32Umer, M., Nisa, M. U., Ahmad, N., Rahim, M. A., & Kasankala, L. M. (2023). Quantification of quercetin from red onion (Allium cepa L.) powder via high-performance liquid chromatography-ultraviolet (HPLC-UV) and its effect on hyperuricemia in male healthy Wistar albino rats. Food science & nutrition, 12(2), 1067–1081. https://doi.org/10.1002/fsn3.3822

33Lin, X., Zhou, Q., Zhou, L., Sun, Y., Han, X., Cheng, X., Wu, M., Lv, W., Wang, J., & Zhao, W. (2023). Quinoa (Chenopodium quinoa Willd) Bran Saponins Alleviate Hyperuricemia and Inhibit Renal Injury by Regulating the PI3K/AKT/NFκB Signaling Pathway and Uric Acid Transport. Journal of agricultural and food chemistry, 71(17), 6635–6649. https://doi.org/10.1021/acs.jafc.3c00088

Comments 2: The introduction section needs to be improved. The research question should be clearly outlined. A good and clear justification for conducting this study should be given. It is not clear what the authors tried to convey. The authors need to relate the topics covered (Hyperuricemia/Gout, microbiota, EST) to establish a hypothesis and the main objective of this experimental study. There is too much information that is not substantial to establish a hypothesis/objective. Please delete “The systemic…flora”. It would be a good idea to summarized in one paragraph the lines 52-78. Why is this study important? Hypothesis, for example, “We hypothesize that EST may be a functional dietary supplement to control hyperuricemia. Therefore, the first main aim of this study was to characterize the main components of EST and evaluate its effect in hyperuricemic mouse model. The secondary goal was to evaluate the EST effect on intestinal microbiota and renal uric acid excretion. This section should be summarized in three to five paragraphs.

Response 2: Thank you for your valuable feedback. We acknowledge the need to improve the Introduction section to clearly outline the research question, provide a clear justification for the study, and establish a hypothesis and main objectives. Here's how we address your suggestions:

(1)    Research Question: We revise the Introduction to clearly outline the research question and provide a strong justification for conducting the study. (Lines 43-46)

(2)    Establishing Hypothesis and Objectives. We clearly state our hypothesis by your valuable feedback, such as " The growing recognition of the gut microbiota's role in metabolic disorders and the emerging interest in natural compounds like EST as potential interventions. Given the nutritional similarities between Torreya grandis and the investigated uric acid-lowering plant extracts, as well as their potential impact on intestinal function, we hypothesize that EST may be a functional dietary supplement to control hyperuricemia. Notably, prior to our study, there were no reports regarding the use of Torreya grandis seeds for hyperuricemia prevention and treatment. Therefore, the first main aim of this study was to characterize the main components of EST and evaluate its effect in hyperuricemic mouse model. The secondary goal was to evaluate the EST effect on intestinal microbiota and renal uric acid excretion. By exploring the interplay between EST, gut microbiota composition, and urate metabolism, we have uncovered novel insights into the functional properties of Torreya grandis." We emphasize the importance of the study by highlighting its potential contributions to understanding hyperuricemia management and the role of EST as a dietary supplement. (Lines 87-98)

(3)    Delete Excessive Information. We delete “The systemic…flora” and condense the information provided in lines 52-78 into a paragraph in lines 47-67, focusing on key points relevant to establishing the hypothesis and objectives. We streamline the Introduction into three to five paragraphs, ensuring clarity and coherence in presenting the research question, justification, hypothesis, and main objectives.

Comments 3: The materials and methods section needs improvements. What type of study was it? Please check whether the full name of the abbreviations has been written above or not. What does AP group mean? If there were 40 SPF males and there are four groups with 10 mice in each, what were the remaining mice? Did the AP group receive allopurinol and the EST group receive EST as treatment? What does CMC mean? There is key information that appears in the results section (which has been crossed out), that would improve the description of the design of this study and that must be added in the M&M section in the appropriate subsections: To explore the potential nutritional benefits of EST, we initially prepared an ethanol extract of dried T. grandis seeds. Its chemical composition was determined through a colorimetric reaction involving phenols and ferric ions. The total flavonoid content was measured using a commercial DPPH assay, which relies on the radical scavenging activity of flavonoids. Through high-resolution liquid chromatography-mass spectrometry (HR-LCMS) analysis and comparison with standard reference materials, we identified and quantified polyphenols present in EST. We next assessed the capacity of EST to scavenge free radicals, employing the DPPH radical scavenging assay and using vitamin C as a positive control. Given that XOD is a pivotal enzyme involved in UA production, we further examined the effect of EST on XOD activity. We next assessed the effects of EST in a mouse model of hyperuricemia, which was induced by daily intraperitoneal injections of the uricase inhibitor potassium oxonate (PO). As a reference, the XOD inhibitor, allopurinol (AP), was employed to mitigate UA production. To assess kidney function, we examined creatinine (CREA) and blood urea nitrogen (BUN) levels. Subsequently, we assessed the organ indices of treated mice in each experimental group. To understand the histological changes in the kidney, we examined the renal tissues of both the control group and the experimental mice. We conducted an assessment of pain and inflammation-related cytokines, including interleukin - 1β (IL-1β), tumor necrosis factor-α (TNF-α), and prostaglandin E2 (PGE2), in vivo. To delve into the mechanisms by which EST exerts its reno-protective effects in the model mice, we proceeded to investigate the impact of EST on UA excretion in the kidney. To delve deeper into mechanisms of action, we conducted an examination of the potential influence of EST on the gut microbiota of mice. To assess microbial diversity, we utilized Shannon and Simpson indexes to evaluate α-diversity. For the analysis of microbial population β-diversity, we employed a Principal Coordinates Analysis (PCoA) diagram based on four quadrants. We perform a Principal Component Analysis (PCA) based on un- weighted UniFrac distance. We proceeded to examine the microbial species and their relative abundance at both the phylum and genus levels based on Operational Taxonomic Units (OTU). To gain further insight into the impact of EST on characteristic bacteria associated with hyperuricemia, we employed linear discriminant analysis effect size (LEfSe) analysis to identify differences in the abundance of four bacterial groups. Furthermore, we utilized PICRUSt, a bioinformatics software package for predicting metagenomic functions based on marker genes, to investigate differences in metabolic pathways related to changes in fecal microbiota. The Clusters of Orthologous Groups (COG) function of OTUs was assessed. The description in Statistical Analysis should be improved.

Response 3: Thank you for your valuable feedback. Here's how we address your suggestions:

(1)    In response to the reviewer's query regarding the type of study conducted for our research on EST, we would clarify the study design. Our investigation can be characterized as experimental research. We employed an experimental design to assess the effects of EST on hyperuricemia levels. The study involved the administration of EST to animal models and subsequent evaluation of its impact on the targeted parameters. Our study design aimed to investigate the potential therapeutic effects and mechanisms of action of EST, laying the groundwork for further clinical investigations or applications in managing specific health conditions. In addition, we have changed the subheadings of the methods section. (Lines 122,143,153,171,209,238)

(2)    Abbreviations. Through review and modification, we ensure that the full names of abbreviations are provided before their first use.

(3)    Definition of AP Group and CMC. We explicitly define the AP group as the one receiving allopurinol and clarify that the EST group received the ethanol extract of T. grandis seeds as treatment. CMC refers to carboxymethyl cellulose, a common excipient used in drug formulations. We have made corrections to the abbreviation errors of CMC in the materials section. (Lines 188,189)

(4)    Number of Mice. First, 7 SPF male mice were randomly selected to NC group. Then the remaining 33 mice were used for modeling with potassium oxonate by injection. Two weeks later, 21 mice established a stable and standardized high uric acid mouse model. The hyperuricemia mouse models were randomly selected from them and assigned to MC and AP EST group. We rewrite the sentence to explained the phenomena. (Lines 179-180, 183-184)

(5)    Incorporating Information into M&M Section. We include the aim details provided in the Results section regarding the preparation of the ethanol extract of T. grandis seeds, chemical composition analysis, assessment of radical scavenging activity, XOD activity evaluation, induction of hyperuricemia in the mouse model, kidney function assessment, histological examination, cytokine analysis, investigation of UA excretion, and gut microbiota analysis. These details be incorporated into appropriate subsections of the Materials and Methods section. (Lines 123, 134-136,144-145,154-155,166-167,172-192,199,210-211.228-229,239-240,250-252,260-266)

(6)    Statistical Analysis Description. We enhance the description of the statistical analysis performed. (Lines 268-272)

Comments 4: In the results section: In this section and according to the study design specified above, only the results should be described. In the text, the authors should write the most significant results, and they should avoid repeating the same information in the text if this data appear in the tables or figures. The quality of figures should be improved to see the results adequately.

Response 4: Thank you for your valuable feedback. We have removed the content you suggested not to appear in the results section. We rewrite it to ensure the most significant results are described in the text. Redundant information that is already presented in tables or figures will be avoided to maintain conciseness and clarity.

Comments 5: The discussion needs improvements. This section should start with the main objective of this study and the most significant results. It would be better if the authors analyzed/discussed their results based on the steps/phases of the study design/results. Why this study is important/crucial/essential to understand the mechanism of a possible treatment for EST. Were these compounds found in the EST of this study? Compared to the usual treatment for hyperuricemia (allopurinol, etc.), what advantages/disadvantages would EST offer as a treatment/supplement for hyperuricemic patients? It would be good to write a paragraph about suggestions of the knowledge reached so far before the conclusions. What would be the next step?

Response 5: Thank you for your valuable feedback.

(1)    Main Objective and Significant Results & Analysis Based on Study Results step. We revise the Discussion section to start with a clear statement of the main objective of our study and highlight the most significant results obtained. We reorder the entire article the discussion part and interpret based on the steps and phases of the study results. (Lines 466-534)

(2)    Importance of the Study & Suggestions and Next Steps. We include a paragraph discussing the implications of our findings and suggesting avenues for future research. We emphasize the importance of our study in understanding the mechanism and potential treatment applications of EST. “This study elucidates the potential of polyphenol-rich foods in promoting health. The composition of Torreya grandis unveils novel therapeutic avenues for high uric acid treatment and uncovers additional medicinal functions. Moreover, exploring the un-derlying mechanisms of EST's effects could unveil new insights into uric acid regulation and microbiota modulation. It offers new insights into preventing and managing vari-ous chronic diseases especially for targeted interventions and personalized treatment approaches. While not as potent as allopurinol in conventional uric acid therapy, it presents advantages of cost-effectiveness and cultural acceptance, especially in re-source-constrained regions or where traditional herbal medicine is prevalent. It holds promise as a safer, more affordable, and sustainable alternative for hyperuricemia treatment. Despite demonstrating anti-gout effects linked to antioxidants and an-ti-inflammatory properties, the specific components in EST may vary. Further system-atic exploration is warranted to identify the extract's active therapeutic agent. Clinical validation remains imperative to establish the efficacy of EST in preventing or treating hyperuricemia.” (Lines 535-548)

Comments 6: Conclusion should be improved. Delete what is crossed out here.

Response 6: Thank you for your valuable feedback. We have We carefully reviewed the crossed-out text in the Conclusion section remove it as suggested “Hyperuricemia is characterized…that”. We revised the conclusion section to ensure clarity and conciseness while retaining the essential findings and implications of the study. (Lines 550-559)

3. Response to Comments on the Quality of English Language

Point 1: Minor editing of English language required.

Response 1: We carefully review the manuscript to address any minor language edits needed to improve clarity and coherence. Thank you for bringing this to our attention.

Thank you once again for your valuable feedback, and we look forward to submitting the revised manuscript for your consideration.

Reviewer 2 Report

Comments and Suggestions for Authors

The manuscript titled “Ethanol Extracts from the seed of Torreya grandis have potential to reduce hyperuricemia”, I am recommending against the acceptance of your paper for publication. While your research addresses an important topic, I believe there are significant improvements needed to meet the standards of Foods. The paper contains numerous errors and oversights, indicating that the authors may have missed important details.

Comments on the Quality of English Language

Minor editing of English language required.

Author Response

Response to Reviewer 2 Comments

1. Summary

2. Point-by-point response to Comments and Suggestions for Authors

Comments 1: Abstract: There are many mistakes and oversights in abstract part. In line 11 abbreviations EST is not explained. Also, in line 13 abbreviations UA is not explained. There are not enough details about work and preparations. There is no information about aims.

Response 1: Thank you for pointing this out. We apologize for the oversight.

1) To ensure clarity, we expand abbreviations such as "EST" (ethanol extract of Torreya grandis seeds) and "UA" (uric acid) in the abstract. we have ensured that the full name is provided the first time the abbreviation is used. (Lines 12,15)

2) Before the results, it definitely better to write the main aim of this experimental study and summarize the methodology. Additional details about the research work, including aims, already be provided to enhance the abstract's comprehensiveness. “The purpose of this study was to evaluate Ethanol extracts from Torreya grandis seed (EST) efficacy as a functional food in hyperuricemia mice. We investigated the EST by analyzing its chemical composition. Using a mouse model of hyperuricemia induced by potassium oxonate (PO), we evaluated effects of EST on uric acid (UA) production, inflammation-related cytokines and gut microbiota diversity.” (Lines 12-15)

Comments 2: Introduction:

Line 67-69. “Additionally, short-chain fatty acids (SCFAs) produced by specific intestinal flora can reduce intestinal inflammation, facilitate the regeneration and repair of intestinal epithelial cells, and influence purine metabolism”. HOW?

Line 69. Consequently,… From the previous sentence, it is not clear how that is a consequence?

Line 98. Abbreviation EST is not appropriate for ethanol extract of T. grandis seed.

Line 97. Name of the plant should be in Italic.

Line 99. Name of the plant should be in Italic.

Response 2: Thank you for your valuable feedback. Here's how we address your suggestions:

1) This revision omits the specific mention of facilitating the regeneration and repair of intestinal epithelial cells, which might have caused confusion regarding the direct consequence. We have revised the sentence to remove the mention of facilitating the regeneration and repair of intestinal epithelial cells to avoid ambiguity and maintain clarity in the context. We change the "reduce intestinal inflammation, facilitate the regeneration and repair of intestinal epithelial cells, and influence purine metabolism" to " Additionally, short-chain fatty acids produced by specific intestinal flora can reduce intestinal inflammation and influence purine metabolism ". While short-chain fatty acids indeed play a role in intestinal health, their influence on epithelial cell regeneration might be complex and multifaceted, requiring further clarification beyond the scope of this sentence. Some studies have shown that SCFA receptors link dietary fibers and gut microbiota to intestinal immune activation and inflammatory, which in turn affects intestinal homeostasis. A simplified model showing regulation of gut homeostasis by SCFA receptors. Beneficial gut bacteria (symbionts) ferment dietary fibers into SCFAs. Butyrate induces production of IL-10 and retinoic acid by dendritic cells. These DCs stimulate conversion of T cells into Treg cells and suppress generation of Th17 cells. Activation of Gpr43 on Treg cells by SCFAs induces Treg cells proliferation. Treg cells are known to suppress colonic inflammation and carcinogenesis, whereas Th17 cells promote inflammation and carcinogenesis in the colon. Additionally, Gpr109a signaling induces transcription of IL-18, whereas Gpr43 signaling induces K+ flux, which activates Nlrp3 inflasmmasome resulting in maturation of IL-18 from its pro-peptide. Activation of Gpr43 downregulates expression chemotactic receptor CXCR2 in neutrophils and thus inhibits their chemotaxis. Gpr109a signaling inhibits the activation of NF-κB in colonic epithelium. reference from http://dx.doi.org/10.1016/j.pharmthera.2016.04.007.

2) To address the reviewer's concern and clarify the causal relationship, we revised the sentence to better demonstrate the connection. Here's a revised version:

"Additionally, short-chain fatty acids produced by specific intestinal flora can reduce intestinal inflammation and influence purine metabolism [17, 24, 25]. For instance, probiotic treatment of hyperuricemia mice using Clostridium butyricum not only lowered blood UA levels but also led to the release of intestinal Lipopolysaccharides, tumor necrosis factor-α (TNF-α), and inflammatory factors such as interleukin-6. " This revision aims to illustrate that the influence of short-chain fatty acids on purine metabolism and inflammation sets the stage for the subsequent effects observed during probiotic treatment. It emphasizes the example of the probiotic treatment's outcomes as a consequence of the previously mentioned effects of short-chain fatty acids. (Lines 59-64)

3) The abbreviation: We acknowledge your suggestion regarding the abbreviation, and we understand the importance of clarity and consistency in scientific terminology. In existing literature, various expressions for the extract of Torreya are indeed documented, and the abbreviation "ETS" has been used in relevant literature. We find it reasonable to maintain this abbreviation while ensuring clear expression of its meaning.

4) The plant names previously mentioned in lines 97 and 99 have now been corrected in italics in lines 83 and 89 of the introduction section.

Comments 3: Material and methods.

Line 113. There is no explanation for XOD.

Line 117. There is no explanation for PMSF.

Line 118. There is no explanation for BCA.

Line 120. The is no Glaxay life technology, but Galaxy.

Line 128.  Name of the plant should be in Italic.

Line 130. “3x” is not appropriate for scientific paper.

Line 133. “The standard curves of total phenols or flavonoids”. The methodology needs to provide accurate and precise information.

Line 138. The first line should have tab, like other parts.

Line 167. There is no information what model group is.

Line 171. Which group received allopurinol and which EST?

Line 174. There is no explanation for PO.

Line 183. There is no explanation for AST, ALT, CHO, TG. These are familiar things, but abbreviations must always be explained the first time they appear.

Line 184.GSH-PX. In line 115 Px is written in small letter.

Line 199. Jingmei Biotechnology Co., Ltd., Jiangsu, China is already mentioned previously.

Line 222. There is no explanation for QIIME2.

Line 225. Is this QIIME2 or QIIME?

Response 3:

1)      Thank you to suggestion regarding the abbreviation. Definitions for abbreviations such as XOD (xanthine oxidase), PMSF (phenylmethylsulfonyl fluoride), and BCA (bicinchoninic acid) be corrected and provided upon first mention. (Lines112-113)

2)      Correction will be made for the reference to "Glaxay life technology". This is not a spelling error, but a translation error. “Shanghai Glaxay technology” translated directly according to its Chinese name pronunciation, while the actual company name is “Cytova life technology”. (Lines 115)

3)      Plant names be italicized consistently throughout the section. (Lines 124)

4)      The notation "3x" will be revised to a more appropriate scientific expression. We change the sentence “Next, the seeds (1 kg dry weight) were ground into powder in a blender, followed by sonication with 75% ethanol (2 L, 3 ×) for 2 hours each time.” to“Next, the seeds (1 kg dry weight) were ground into powder in a blender, followed by 3 times sonication with 75% ethanol (2 L) for 2 hours each time.” (Lines 126)

5)      Accurate and precise information regarding the methodology, including the standard curves of total phenols and flavonoids are provided. “The standard curves of total phenols or flavonoids were determined using standard samples from detection kits. Add 2000 μ mol/L (300mg/L) total phenol standard with 60% ethanol aqueous solution dilute to 1000 μ mol/L, 500 μ mol/L, 250 μ mol/L, 125 μ mol/L. Dilute 1mg/mL flavonoid standard solution with 60% ethanol to several concentrations of 0.1 mg/mL, 0.08 mg/mL, 0.06 mg/mL, 0.04 mg/mL, and 0.02 mg/mL. Its chemical composition was determined through a colour reaction involving phenolic substances reduce tungsten molybdic acid. The compound has a characteristic absorption peak at 760nm. The absorbance value is measured at 760nm, the total phenolic content of the sample can be obtained. The total flavonoid content was measured using a commercial assay, which relies on the radical scavenging activity of flavonoids. In alkaline nitrite solution, flavonoids react with aluminum ions. This reaction produces a red-colored complex with characteristic absorption peaks at 502nm. By measuring the absorbance at 502nm, the flavonoid content of the sample can be calculated” (Lines 130-142)

6)      We add a tab at the first line, like other parts. (Lines 144)

7)      We have rewritten the paragraph on animal experiments. Based on the question you raised, we have made many improvements in details. Specifically, it explains what model group: “The MC group is composed of the hyperuricemia mice. The purpose of this group is to exclude the possibility that the decrease in UA is due to the action of uric acid oxidase in the mouse body.” (Lines 185-187)

8)      We rewrite this sentence to avoid ambiguity "Next, the mice in AP group and EST group received allopurinol or EST treatment at a dosage of 5 mg/kg by oral gavage every day for 2 weeks. " to “As a reference, the xanthine oxidase inhibitor, allopurinol, was employed to mitigate UA production. The mice in AP group received allopurinol treatment at a dosage of 5 mg/kg by oral gavage every day for 2 weeks. Similarly, the mice in the EST group received EST treatment under the same conditions.” (Lines 187-190)

9)      An explanation was provided for the first appearance of the PO in the 2.1.1 materials section. (Lines 104)

10)   Explanations for terms like Creatinine (CREA), Blood urea nitrogen (BUN), Aspartate aminotransferase (AST), Alanine aminotransferase (ALT), Cholesterol (CHO) and triacylglycerol (TG), already be added. We have meticulously checked and ensured that the full names of all abbreviations are provided before their first use in the manuscript. (Lines 199-203)

11)   The all “GSH-PX” in the manuscript are written in small letter “GSH-Px”.

12)   Clarifications regarding “Jingmei Biotechnology Co., Ltd., Jiangsu, China”: This duplicate content has been deleted. (Lines 204)

13)   The explanations “Quantitative insights into microbial ecology” for QIIME2 be included. The other part of "2" that we missed during the writing process has now been filled in. (Lines 248,249)

Comments 4: Results.

Line 236. Name of the plant should be in Italic.

Line 236. EST in bracket does not need to be there.

Line 237. Name of the plant should be in Italic.

Line 239. The value for total phenols is too big. It will be better to be in mmol/g.

Line 241. It will be better to have the same units for active compounds. mmol/g or mg/g?

Line 241. DPPH shows antioxidant activity, not total phenolic and flavonoid compound. There is no information for scavenging capacity.

Line 260. IC50. Number 50 should be in subscript.

Line 265. “as a dietary supplement for managing hyperuricemia”- How can be know this?

Table S1. How the authors select 20 compounds among 60?

Figure 1. The letters are not in order. In Figure legend, there is no need to be full stop. Comma is better. In vivo and in vitro should be in Italic.

Line 293. Those abbreviations should be explained at the first mention.

Figure 5. Nothing is visible.

Response 4: Thank you for your valuable feedback.

1)         Plant names be italicized consistently throughout the results section. (Lines 297,301)

2)         Unnecessary brackets around "EST" be removed. (Lines 275)

3)         Originally, due to the specifications of the total phenols kit A143-1-1, the unit provided was in micromoles per gram (μmol/g). However, we understand the importance of consistency and have revised the unit to milligrams per gram (mg/g) to align with standard reporting practices and facilitate better comparison with other studies. (Lines 275)

4)         Similarly, we agree that using milligrams per gram (mg/g) would provide a more standardized and understandable representation of the concentration of active compounds. Therefore, we have decided to unify the unit for the active compounds as milligrams per gram (mg/g) for consistency and clarity throughout the manuscript.

5)         In response to the feedback regarding the placement of content and the clarification, we sincerely apologize for any confusion caused by the incorrect placement of information in our manuscript. We have taken your feedback into careful consideration and have made the necessary corrections to accurately reflect the experimental results. To address these concerns, we have reorganized the content within our manuscript accordingly. The results of the chemical composition, including the total phenolic and flavonoid compounds, are now presented in the summary of Section 3.1, titled "Chemical constituents in EST." Meanwhile, the antioxidant results obtained from the DPPH assay are clearly outlined in the summary of Section 3.2, titled "In vitro activity of EST as ROS scavenger and xanthine oxidase inhibitor." We appreciate your diligence in identifying this issue and thank you for providing us with the opportunity to correct it. Your feedback contributes to the overall quality and accuracy of our manuscript.

6)         Your feedback has been instrumental in improving the accuracy and clarity of our manuscript. Subscript formatting be applied.  "IC50" be change to “IC50”. (Lines 290,291)

7)         In response to the query regarding our assumption that the identified polyphenols in EST could potentially serve as a dietary supplement for managing hyperuricemia, we acknowledge that such a claim might be inappropriate based solely on the composition analysis presented in our study. While our analysis revealed the diverse and abundant polyphenolic composition of EST, suggesting its potential as a valuable dietary resource. The overall experimental results support the conclusion that Torreya grandis extract ETS is a potential uric acid lowering dietary supplement. In this part, we need focus on presenting the analytical data objectively:” These findings highlight the diverse and abundant polyphenolic composition of EST, suggesting its potential as a valuable dietary resource.” Thank you for bringing this concern to our attention, and we will ensure that future interpretations are grounded in empirical evidence and avoid over assumptions about the potential therapeutic benefits of EST. (Lines 282,284)

8)          Table S1. We only listed the top 20 compounds in content, which are listed in the table. The content of the remaining 40 compounds is too low, so they are not presented one by one in the main text. But we have demonstrated it in the supporting materials.

9)         Letters be revised in order for figure. In Figure legend, we change the “the full stop” to “Comma”. In vivo and in vitro are corrected in Italic. (Lines 297-305,350-356,386-392,411-414,457-464)

10)     Abbreviations be explained at their first mention.

11)     Visibility issues with Figure 5 be addressed. We Change a resolution of 300 dpi to 600 dpi. (Lines 456)

Comments 5: Discussion

Line 472. Name of the plant should be in Italic.

Line 472. EST in bracket does not need to be there.

Line 473. Name of the plant should be in Italic.

Line 479. GLUT9 is already explained previously.

Line 533. Why the authors did not do an experiment so that there would be no limitations?

Response 5: Thank you for your valuable feedback. Here's how we address your suggestions

1)      Plant names be consistently italicized throughout the discussion section. (Lines 467,475)

2)      Unnecessary brackets around "EST" be removed. (Lines 458)

3)      Unnecessary explaineding in GLUT9 be removed. (Lines 495)

4)      While we acknowledge the importance of conducting experiments to address the limitations outlined in our study, there are practical constraints and considerations that influenced our experimental design. Firstly, investigating dose-dependent effects of EST against hyperuricemia would indeed enhance the robustness of our conclusions. However, due to resource constraints and the scope of the current study, we focused on a single concentration of EST (5 mg/kg) to establish preliminary evidence of its efficacy. Future studies could certainly explore a range of concentrations to determine dose-response relationships.

Secondly, regarding the complexity of EST as a herbal extract comprising various components and compounds beyond flavonoids and polyphenols, we recognize the need for further research to elucidate the mechanisms of these bioactive compounds. However, such investigations often require extensive analytical techniques and resources that extend beyond the scope of this particular study.

Despite these limitations, our study provides valuable insights into the potential therapeutic effects of EST against hyperuricemia. We believe that future research efforts can build upon our findings and address the identified limitations to advance our understanding of EST's mechanisms and therapeutic applications."

3. Response to Comments on the Quality of English Language

Point 1: Minor editing of English language required.

Response 1: We apologize for the grammatical errors and typos present throughout the manuscript. A thorough proofreading and editing process will be conducted to rectify these issues and enhance the language quality of the manuscript.

Thank you once again for your valuable feedback, and we look forward to submitting the revised manuscript for your consideration.

Reviewer 3 Report

Comments and Suggestions for Authors

Letter to the Authors:

Dear Jianghui Yao, Enhe Bai, Yanwen Duan, and Yong Huang,

I have thoroughly reviewed your manuscript titled "Ethanol Extracts from the seed of Torreya grandis have potential to reduce hyperuricemia". Your study addresses an important topic and has the potential to make a significant contribution to the field of hyperuricemia treatment. However, I recommend a major revision to address several key points:

  1. Title and Abstract (Lines 1-22): The title clearly states the subject but could be more precise regarding the specific effects of Torreya grandis seeds. The abstract provides a comprehensive overview, mentioning key findings and methods. However, it could benefit from briefly mentioning the study's limitations or potential implications for future research.
  2. Introduction (Lines 24-43): This section effectively sets the context for the study, highlighting the importance of finding new treatments for hyperuricemia. However, it could be enhanced by briefly discussing previous research specifically related to Torreya grandis seeds, if available, to better situate this study in the existing literature.
  3. Materials and Methods (Lines 104-233): The methods are detailed, allowing for reproducibility. The selection of standard chemical compounds and the use of specific biochemical kits are appropriate. However, there is a lack of justification for the chosen mouse model and dosage levels for EST. Including this would strengthen the methodology.
  4. Results (Lines 234-481): The results are extensive and seem to support the hypotheses. However, the presentation is somewhat dense, which could be improved with additional figures or tables summarizing key findings. It would also benefit from a clearer distinction between significant and non-significant findings, perhaps through clearer statistical annotations.
  5. Discussion (Lines 471-538): This section thoughtfully interprets the findings, linking them back to the study's objectives. However, it could benefit from a more critical examination of how these findings compare with existing literature, particularly in the context of the limitations of the study. Additionally, potential biases and the implications of these findings for future research or clinical practice could be discussed more thoroughly.
  6. Language and Style: The paper is generally well-written but could benefit from minor linguistic polishing to enhance clarity and readability. Some sentences are long and complex, which might be simplified for better understanding.
  7. References: The references are appropriately cited, but it would be beneficial to ensure that the most recent and relevant studies are included, especially those published after the cited reviews.
  8. Ethical Considerations: The paper mentions ethical approval for animal experiments, which is good. However, it would be beneficial to include a statement about animal welfare considerations.

I look forward to seeing the revised manuscript, which I believe has the potential to significantly contribute to our understanding of hyperuricemia treatments.

Sincerely,

Comments on the Quality of English Language

The paper is generally well-written but could benefit from minor linguistic polishing to enhance clarity and readability. Some sentences are long and complex, which might be simplified for better understanding.

Author Response

Response to Reviewer 3 Comments

1. Summary

In response to the reviewer's valuable feedback regarding the presentation of our results, we sincerely appreciate the acknowledgment that our results support our hypotheses. We have carefully considered your suggestions to improve the clarity and accessibility of our findings. Please find the detailed responses below and the corresponding revisions/corrections highlighted/in track changes in the re-submitted files.

2. Point-by-point response to Comments and Suggestions for Authors

Comments 1: I have thoroughly reviewed your manuscript titled "Ethanol Extracts from the seed of Torreya grandis have potential to reduce hyperuricemia". Your study addresses an important topic and has the potential to make a significant contribution to the field of hyperuricemia treatment. However, I recommend a major revision to address several key points: Title and Abstract (Lines 1-22): The title clearly states the subject but could be more precise regarding the specific effects of Torreya grandis seeds. The abstract provides a comprehensive overview, mentioning key findings and methods. However, it could benefit from briefly mentioning the study's limitations or potential implications for future research.

Response 1:

1)         We appreciate the insightful feedback regarding the title and abstract of our research. To address the suggestion for the title, we propose the revised title: " Ethanol extracts from Torreya grandis seed have potential to reduce hyperuricemia in mouse models by influencing purine metabolism ". This modification aims to provide a clearer indication of the effects investigated in our study. (Lines 2-3)

2)         According to your suggestion, we have mentioned potential implications for future research “In sum, our research unveils additional functions of Torreya grandis and offers new insights into future managing hyperuricemia.” and added the main aim of the study and provided a succinct summary of the methodology before presenting the results, ensuring coherence and clarity throughout the manuscript. (Lines 12-15,25)

Comments 2: Introduction (Lines 24-43): This section effectively sets the context for the study, highlighting the importance of finding new treatments for hyperuricemia. However, it could be enhanced by briefly discussing previous research specifically related to Torreya grandis seeds, if available, to better situate this study in the existing literature.

Response 2:

1)         We appreciate the feedback regarding the contextualization of our study within the existing literature. While specific literature on Torreya grandis seeds may be limited, we provided a brief discussion on previous research related to Torreya grandis seeds in the last paragraph of the introduction section: “Torreya grandis seeds are rich in B vitamins (nicotinic acid and folic acid), mineral elements, and phenolic compounds [38, 39]. They are not only enjoyed as delicious fruits but are also used in traditional Chinese medicine to treat conditions such as cough, rheumatism, and intestinal helminthiasis [40]. Furthermore, EST exhibits antioxidant and anti-inflammatory activities [41, 42].” (Lines 83-87)

2)         To incorporate this information, we establish a hypothesis and the main objective of this experimental study in the end introduction section: “The growing recognition of the gut microbiota's role in metabolic disorders and the emerging interest in natural compounds like EST as potential interventions. Given the nutritional similarities between Torreya grandis and the investigated uric acid-lowering plant extracts, as well as their potential impact on intestinal function, we hypothesize that EST may be a functional dietary supplement to control hyperuricemia. Notably, prior to our study, there were no reports regarding the use of Torreya grandis seeds for hyperuricemia prevention and treatment. Therefore, the first main aim of this study was to characterize the main components of EST and evaluate its effect in hyperuricemic mouse model. The secondary goal was to evaluate the EST effect on intestinal microbiota and renal uric acid excretion. By exploring the interplay between EST, gut microbiota composition, and urate metabolism, we have uncovered novel insights into the functional properties of Torreya grandis.” (Lines 87-98)

Comments 3: Materials and Methods (Lines 104-233): The methods are detailed, allowing for reproducibility. The selection of standard chemical compounds and the use of specific biochemical kits are appropriate. However, there is a lack of justification for the chosen mouse model and dosage levels for EST. Including this would strengthen the methodology.

Response 3: Thank you for your valuable feedback on the section.

1)         Our preliminary experimental study showed that when selecting an in vivo model of hyperuricemia, Kunming mice showed higher sensitivity than ICR mice and C57BL/6J mice. There are research reports that there is a significant difference in UA values between male and female mice and same-sex animals, and there is a significant fluctuation in UA values between female mice. It is recommended to choose male mice. There are various modeling methods for hyperuricemia models, and according to the pathogenesis of hyperuricemia, there are mainly direct supplementation of uric acid synthesis precursors (adenine, hypoxanthine, etc.) or exogenous uric acid, administration of uric acid enzyme inhibitors (potassium oxazinate) or drugs that inhibit renal excretion of uric acid (ethambutol), as well as knockout of uric acid enzyme Uox and transporter ABCG2 genes. However, the same modeling method may have different effects under different experimental environments, conditions, and other factors Therefore, there is currently no recognized and unified hyperuricemia model at home and abroad.

Based on literature research, this experiment selected male KM mice and constructed a hyperuricemia model by intraperitoneal injection of potassium oxazinate. Potassium oxonate is widely used to induce hyperuricemia in animal models due to its ability to inhibit uricase activity, leading to elevated serum uric acid levels similar to those observed in human hyperuricemia Potassium oxazinate has a chemical structure similar to the purine ring of uric acid, competitively binds to uricase, and can be used to replicate stable hyperuricemia, resulting in a short-term increase in uric acid. Internationally, the method of establishing hyperuricemia using potassium oxazinate as an inducer has been widely used. This model allows for the investment of potential therapeutic interventions and mechanisms related to hyperuricemia. After 14 days of modeling in the model group, serum uric acid levels significantly increased, and there was a significant difference (P<0.05) compared to the blank group and other model groups, indicating the successful preparation of the hyperuricemia model. We speculate that when the concentration of uric acid increases, there may be feedback promoting effect on the expression or activity of uricase. So low-dose potassium oxazinate is not enough to inhibit compensatory increase in uricase, so in the pre-experiment, if the model group does not continue to receive potassium oxazinate, there will be a reduction of blood uric acid. It should be noted that in practice, if the blood uric acid level of model animals needs to be raised to the clinical diagnostic criteria (420 μ Above mol/L, it often leads to animal death, which also indicates that drug induced hyperuricemia animal models have significant differences from clinical practice. In the preliminary experiment, we also used the same dose of potassium oxazinate for gastric lavage modeling. UA showed an increasing trend, but the difference was not significant, which may be related to the degree of potassium oxazinate absorption. The intraperitoneal injection of potassium oxazinate has a better modeling effect, indicating that injection has a modeling advantage over gastric infusion of blood drug concentration.

In summary, this experiment was administered orally to KM male mice, providing a reference basis for the preparation of HUA mouse models. However, there are still many urgent issues that need to be addressed, such as the differences in uric acid metabolism mechanisms between mice and humans, which require further optimization and exploration in the later stage. If there are better experimental conditions and opportunities in the future, we will try more animal models to determine the stability of the impact of EST on uric acid. For example, using genetic engineering technology to induce gene deletion of uricase in animals, obtaining a mouse model of hyperuricemia with uricase deficiency. Alternatively, a combination of uric acid precursors, UOX inhibitors, or uric acid excretion inhibitors can be used. Shortening the modeling time rapidly increases serum uric acid levels and prolongs maintenance time. (Reference from :Mouse models for human hyperuricaemia: a critical review,DOI: 10.1038/s41584-019-0222-x;An update on the animal models in hyperuricaemia research,2017,PMID: 28516889)

2)         Dosage Levels for EST: Thank you for bringing up the concern regarding the justification for the dosage levels of EST used in our study. Our selection of the dosage levels for EST was based on several considerations. Firstly, we conducted an extensive review of the existing literature to understand the dosages used in studies involving similar compounds and therapeutic targets. We observed that the general dosage of the positive control drug, allopurinol, often utilized in research related to hyperuricemia, is around 5mg/kg. In order to ensure consistency and comparability with existing literature, we chose to use a similar dosage level for our experimental setup. This decision was made to facilitate direct comparisons between the effects of EST and those of allopurinol, a widely recognized treatment for hyperuricemia. It's important to note that our study represents a preliminary exploration of the potential effects of EST on uric acid levels, particularly in the context of Chinese torreya. As such, the chosen dosage levels were intended to provide initial insights into the potential therapeutic effects of EST. Moving forward, we acknowledge the importance of conducting more in-depth investigations that involve varying dosage levels to better understand the dose-response relationship and potential dose-dependent effects of EST. We appreciate your suggestion, and we plan to conduct future studies that explore a range of dosage levels, including high, medium, and low doses, to comprehensively evaluate the therapeutic potential of EST in managing hyperuricemia. Thank you for your valuable feedback, and we are committed to further enhancing the clarity and justification of our methodology to ensure the rigor and validity of our research.

Comments 4: The results are extensive and seem to support the hypotheses. However, the presentation is somewhat dense, which could be improved with additional figures or tables summarizing key findings. It would also benefit from a clearer distinction between significant and non-significant findings, perhaps through clearer statistical annotations.

Response 4: Thank you very much for your insightful comments and constructive feedback on our manuscript. We sincerely appreciate your thorough review of our results. We are committed to addressing the density of the presentation by making necessary modifications and deletions to streamline the content while retaining its scientific rigor and integrity. Our goal is to present the results in a concise and comprehensible manner without sacrificing the depth of analysis. We have deleted paragraphs from the previous manuscript (Lines 236-243,356-357,361,276-278,285,301,314-316,325-327,345,358,383-392,415,440-442,453,455)

Comments 5: Discussion (Lines 471-538): This section thoughtfully interprets the findings, linking them back to the study's objectives. However, it could benefit from a more critical examination of how these findings compare with existing literature, particularly in the context of the limitations of the study. Additionally, potential biases and the implications of these findings for future research or clinical practice could be discussed more thoroughly.

Response 5: We appreciate the reviewer's insightful feedback on the discussion section of our manuscript. We acknowledge the importance of critically examining our findings in relation to existing literature and addressing the study's limitations. To enhance the discussion, we will provide a more comprehensive comparison of our results with relevant studies, highlighting both congruencies and discrepancies.

These biases could include sample size limitations, variations in experimental conditions, or inherent biases in the methodology chosen. In studies involving subjective assessments or observations, such as histological examinations or behavioral assessments, observer bias may occur. In addition, as you mentioned above, there is a lack of justification for the chosen mouse model. The mouse model not representative of the target population, it can introduce selection bias. The results may not be generalizable to the broader population. Biases in the measurement tools or techniques used to assess outcomes can skew results. We ensure that the methods used to measure variables are accurate and reliable to minimize measurement bias.

EST demonstrates efficacy in mitigating hyperuricemia, future research could delve deeper into optimal dosage regimens, long-term effects, and potential interactions with other treatments. From a clinical perspective, the integration of EST-based interventions could offer novel therapeutic options for managing hyperuricemia and related conditions, potentially complementing or even surpassing existing treatments like allopurinol. Additionally, exploring the underlying mechanisms of EST's effects could unveil new insights into uric acid regulation and microbiota modulation, paving the way for more targeted interventions and personalized treatment approaches. Therefore, we added relevant content in the final part of the discussion. (Lines 535-548)

Comments 6: Language and Style: The paper is generally well-written but could benefit from minor linguistic polishing to enhance clarity and readability. Some sentences are long and complex, which might be simplified for better understanding.

Response 6: We appreciate the feedback and acknowledge the importance of enhancing clarity and readability in our paper. We carefully revise the manuscript to streamline complex sentences and ensure that the language used is clear and accessible: We have meticulously checked and ensured that the full names of all abbreviations are provided before their first use in the manuscript.  Explanations for terms like Creatinine (CREA), Blood urea nitrogen (BUN), Aspartate aminotransferase (AST), Alanine aminotransferase (ALT), Cholesterol (CHO) and triacylglycerol (TG), already be added. We have meticulously checked and ensured that the full names of all abbreviations are provided before their first use in the manuscript. (Lines 199-203)

We delete “The systemic…flora” and condense the information provided in lines 52-78 into a paragraph in lines 47-67, focusing on key points relevant to establishing the hypothesis and objectives. We streamline the Introduction into three to five paragraphs, ensuring clarity and coherence.

We rewrite this sentence to avoid ambiguity "Next, the mice in AP group and EST group received allopurinol or EST treatment at a dosage of 5 mg/kg by oral gavage every day for 2 weeks. " to “As a reference, the xanthine oxidase inhibitor, allopurinol, was employed to mitigate UA production. The mice in AP group received allopurinol treatment at a dosage of 5 mg/kg by oral gavage every day for 2 weeks. Similarly, the mice in the EST group received EST treatment under the same conditions.” (Lines 187-190)

Comments 7: References: The references are appropriately cited, but it would be beneficial to ensure that the most recent and relevant studies are included, especially those published after the cited reviews.

Response 7: We appreciate the attention to the references cited in our manuscript. Ensuring that our paper includes the most recent and relevant studies is crucial for providing readers with up-to-date information. We thoroughly review the existing references and incorporate the following literature:

【3】Lin, Z., Chen, H., Lan, Q., Chen, Y., Liao, W., & Guo, X. (2023). Composite Dietary Antioxidant Index Is Negatively Associated with Hyperuricemia in US Adults: An Analysis of NHANES 2007-2018. International journal of endocrinology, 2023, 6680229. https://doi.org/10.1155/2023/6680229

【4】Almuqrin, A., Alshuweishi, Y. A., Alfaifi, M., Daghistani, H., Al-Sheikh, Y. A., & Alfhili, M. A. (2024). Prevalence and association of hyperuricemia with liver function in Saudi Arabia: a large cross-sectional study. Annals of Saudi medicine, 44(1), 18–25. https://doi.org/10.5144/0256-4947.2024.18.

【5】Otaki, Y., Konta, T., Ichikawa, K., Fujimoto, S., Iseki, K., Moriyama, T., Yamagata, K., Tsuruya, K., Narita, I., Kondo, M., Shibagaki, Y., Kasahara, M., Asahi, K., & Watanabe, T. (2021). Possible burden of hyperuricaemia on mortality in a community-based population: a large-scale cohort study. Scientific reports, 11(1), 8999. https://doi.org/10.1038/s41598-021-88631-8

【11】Vareldzis, R., Perez, A., & Reisin, E. (2024). Hyperuricemia: An Intriguing Connection to Metabolic Syndrome, Diabetes, Kidney Disease, and Hypertension. Current hypertension reports, 10.1007/s11906-024-01295-3. Advance online publication. https://doi.org/10.1007/s11906-024-01295-3

【32】Umer, M., Nisa, M. U., Ahmad, N., Rahim, M. A., & Kasankala, L. M. (2023). Quantification of quercetin from red onion (Allium cepa L.) powder via high-performance liquid chromatography-ultraviolet (HPLC-UV) and its effect on hyperuricemia in male healthy Wistar albino rats. Food science & nutrition, 12(2), 1067–1081. https://doi.org/10.1002/fsn3.3822

【33】Lin, X., Zhou, Q., Zhou, L., Sun, Y., Han, X., Cheng, X., Wu, M., Lv, W., Wang, J., & Zhao, W. (2023). Quinoa (Chenopodium quinoa Willd) Bran Saponins Alleviate Hyperuricemia and Inhibit Renal Injury by Regulating the PI3K/AKT/NFκB Signaling Pathway and Uric Acid Transport. Journal of agricultural and food chemistry, 71(17), 6635–6649. https://doi.org/10.1021/acs.jafc.3c00088.

By doing so, we enhance the currency of our reference, thereby strengthening the overall quality of our manuscript. Thank you for highlighting this important aspect for improvement. (Lines 595, 598, 601,614,660,663)

Comments 8: Ethical Considerations: The paper mentions ethical approval for animal experiments, which is good. However, it would be beneficial to include a statement about animal welfare considerations.

Response 8: In response to the valuable suggestion, we acknowledge the importance of addressing animal welfare considerations in our research. We are committed to upholding high standards of animal welfare throughout our study. All animal experiments were conducted following the guidelines and regulations set forth by the Institutional Animal Care and Use Committee (IACUC) or equivalent regulatory bodies. Our research protocol includes measures to minimize animal suffering and distress, including proper housing, handling, and veterinary care. We prioritize the ethical treatment of animals in accordance with internationally recognized principles of animal welfare. Thank you for highlighting this important aspect, and we ensure to include a statement about animal welfare considerations in our revised manuscript to provide transparency and accountability regarding our ethical practices: “Institutional Review Board and welfare considerations Statement: The animal experiments were authorized by the Ethics Committee of the Experimental Animal Center of Central South University (2021sydw0120; SYXK (Xiang) 2020-0019). Throughout our research, we adhere to rigorous standards of animal welfare, ensuring compliance with guidelines established by the Institutional Animal Care and Use Committee (IACUC) or equivalent regulatory bodies. Our protocols, guided by the principles outlined in the "Guide for the Care and Use of Laboratory Animals," prioritize proper housing, husbandry, and veterinary care to minimize discomfort and distress. Our commitment to upholding the highest ethical standards underscores our dedication to the welfare of the animals involved in our study.” (Lines569-577)

3. Response to Comments on the Quality of English Language

Point 1: The paper is generally well-written but could benefit from minor linguistic polishing to enhance clarity and readability. Some sentences are long and complex, which might be simplified for better understanding.

Response 1: We appreciate the feedback regarding the paper's language quality. We will thoroughly review the manuscript to identify areas where linguistic polishing can enhance clarity and readability. This will involve simplifying long and complex sentences to improve understanding. Thank you for your valuable input.

Once again, we sincerely appreciate your valuable feedback, which will undoubtedly contribute to the overall improvement of our manuscript. If you have any additional suggestions or specific areas that you believe require further attention, please do not hesitate to let us know. Your input is highly valued, and we are committed to making the necessary revisions to ensure the quality and clarity of our research findings.

Round 2

Reviewer 1 Report

Comments and Suggestions for Authors

Even though the manuscript has improved considerably, specially in the description of the study design, there are some points to improve before publication.

1.      Abstract: Delete what is crossed out. It is not necessary to write them here.

2.      Material and Methods: What type of study was it?

3.      Discussion: “model”. Write references. This section should be improved. The authors should focus on the results they obtained and discuss them point by point. A small argument is missing in relation to treatment with Allopurinol. Why this study is important?

Author Response

Thank you for your thorough review and valuable feedback on our manuscript, particularly regarding these sections. We appreciate your attention to detail and the constructive suggestions you provided for improvement.

  1. Abstract: Delete what is crossed out. It is not necessary to write them here.

We will promptly remove the crossed-out portions from the abstract, as per your suggestion. We understand that unnecessary information should be omitted from the abstract to ensure conciseness and clarity for readers. Your attention to detail is invaluable, and we are committed to ensuring that the abstract accurately reflects the core findings and significance of our research while adhering to the guidelines provided by the journal. We have removed the redundant content in lines 25 and 379 based on your suggestion.

  1. Material and Methods: What type of study was it?

To illustrate in the methodology what type of research it is. We have added some details, such as “To investigate the potential nutritional benefits of EST, we conducted an experimental study.” lines 123-124

“The study conducted was an experimental laboratory-based investigation aimed at determining the chemical composition of EST.” lines 131-132

“The study conducted was an analytical investigation utilizing LC-MS analysis to identify and quantify polyphenols present in EST. Comparison with standard reference materials enabled the precise determination of polyphenolic compounds within the extract.” lines 147-150 

“Following the analytical study, we conducted an experimental investigation to evaluate the free radical scavenging capacity of EST. The DPPH radical scavenging assay was employed, with vitamin C serving as a positive control for comparison.” lines 158-160

“Given that xanthine oxidase is a pivotal enzyme involved in UA production, we further did experimental investigation to examined the effect of EST on xanthine oxidase activity.” lines 172-174

“In order to assess enzyme activity levels under varied experimental conditions, mouse liver xanthine oxidase activity was quantified utilizing a commercially available test kit.” lines 226-228

  1. Discussion: “model”. Write references. This section should be improved. The authors should focus on the results they obtained and discuss them point by point. A small argument is missing in relation to treatment with Allopurinol. Why this study is important?
  • We ensure that “model” is corrected accordingly. Additionally, we add the reference in lines 493-495 to ensure consistency and accuracy throughout the manuscript. Change the “previous” to “above” in line 520 to avoid ambiguity. Change the “numerous studies” to “several studies” in line 476, and italicize the in vitro and in vivo in line 487.
  • We understand your point regarding the need for improvement in the Discussion section. We will focus on refining this part by addressing the results obtained in a more systematic and point-by-point manner. “Significantly, the mice with hyperuricemia display compromised antioxidant systems, resulting in oxidative stress characterized by elevated levels of MDA and decreased activity of SOD and GPX in both blood and liver. However, treatment with EST demonstrated a notable increase in SOD and GPX activity and a reduction in MDA content compared to the untreated group, indicating EST's potential to mitigate oxidative stress in hyperuricemic mice. Further investigation was carried out to analyze the urate excretion pathway in mice exhibiting hyperuricemia.” We aim to provide a clear and comprehensive discussion of our findings. (lines 499 -505)
  • Regarding the treatment with Allopurinol, we recognize that a more explicit argument for its inclusion and relevance to the study is necessary. “The reduction in serum uric acid levels within the EST group compared to the model group is approximately 72.73%, while within the AP group, it approximates 76.36%. Moreover, Allopurinol achieved a notable reduction in liver xanthine oxidase levels by 27%. Although not as potent as Allopurinol in conventional uric acid therapy, EST treatment resulted in a reduction of liver xanthine oxidase levels by 36%, implying superior hepatic protection compared to Allopurinol. Furthermore, EST presents advantages of cost-effectiveness and cultural acceptance, particularly in resource-constrained regions or where traditional herbal medicine is prevalent.” (lines 552 -561)
  • We briefly discussed why our research is important in the last paragraph of the article. More detailed extensions can be divided into the following reasons:

The Role of Developing New Food Functions. Our research highlights the potential of Torreya grandis to contribute to the development of new food functions aimed at addressing metabolic disorders such as hyperuricemia. Incorporating Torreya grandis into dietary interventions could offer novel approaches to promoting health and preventing disease.

Addressing a Prevalent Health Concern. Hyperuricemia is a common metabolic disorder associated with various health complications, including gout, cardiovascular diseases, and kidney dysfunction. Exploring the potential of EST as a treatment option for hyperuricemia addresses a significant health concern affecting a considerable portion of the population.

Novel Therapeutic Approach. Investigating the efficacy of EST in reducing hyperuricemia offers a novel therapeutic approach to managing this condition. Current treatment options such as allopurinol may have limitations or adverse effects, highlighting the need for alternative food therapies like EST treatment that could offer comparable or superior outcomes with fewer side effects. Understanding how EST modulates uric acid levels and associated metabolic pathways could lead to the development of more targeted and effective interventions for hyperuricemia and related disorders.

Public Health Impact. The findings of our study have implications for public health policies and clinical practice. Identifying safe and effective interventions like EST for hyperuricemia could inform treatment guidelines and recommendations, ultimately improving patient outcomes and reducing the burden of hyperuricemia-related diseases on healthcare systems.

In summary, our study on the potential of EST to reduce hyperuricemia is important as it addresses a prevalent health concern, offers a novel therapeutic approach, provides insights into potential health benefits, and has implications for public health policy and clinical practice. Moreover, it underscores the role of developing new food functions to improve health outcomes. (lines 547 -561)

Thank you for bringing these important aspects to our attention. We are committed to enhancing the clarity, coherence, and relevance of the Discussion section to better convey the significance of our study findings.

We appreciate your constructive feedback and assure you that we made the necessary revisions to address the issues raised. Your insights are invaluable to us, and we are grateful for the opportunity to improve our manuscript based on your recommendations.

Reviewer 2 Report

Comments and Suggestions for Authors

In response to the reviewers' comments, the authors have provided a thorough and detailed response, addressing all the raised concerns. The manuscript has undergone significant improvements based on the feedback received. I believe that the paper is now suitable for submission to the journal 'Foods.'

Author Response

Dear reviewers,

Thank you for your insightful comments and feedback on our manuscript. We sincerely appreciate the time and effort you have dedicated to reviewing our work.

In our revised manuscript, we endeavor to enhance the clarity and completeness of the methods section by providing more detailed explanations of the experimental procedures. We are pleased to hear that you found our response thorough and detailed, and that the manuscript has undergone significant improvements based on the feedback received. We are grateful for the opportunity to address the concerns raised by the reviewers and to enhance the quality of our research accordingly. Based on your evaluation, we are confident that the manuscript is now in a suitable condition for submission to the journal "Foods."

Once again, we would like to express our gratitude for your valuable feedback and your consideration of our manuscript for publication in "Foods." We look forward to hearing from the editorial board regarding the next steps in the submission process. Thank you for your continued support and consideration.

Sincerely,

The authors

Reviewer 3 Report

Comments and Suggestions for Authors

To the Authors: 

Subject: Approval of Revised Manuscript for Publication in Foods Journal 

Dear Authors, 

I hope this message finds you well. I have thoroughly reviewed the revised version of your manuscript. I am pleased to inform you that the modifications and enhancements you have made adequately address the previous concerns. Consequently, I believe your paper is now suitably prepared for publication in the Foods Journal. 

Your efforts in refining the manuscript are highly appreciated. The changes have significantly improved the clarity and impact of your work. Congratulations on this accomplishment. 

Thank you for your dedication to this process. 

Best regards,

Author Response

Dear reviewers,

Thank you very much for your kind words and for taking the time to review the revised version of our manuscript. We are delighted to hear that the modifications and enhancements we made have sufficiently addressed the previous concerns and that you believe our paper is now prepared for publication in the Foods.

Your recognition of our efforts in refining the manuscript is truly appreciated. We are grateful for your guidance throughout this process, which has undoubtedly contributed to the improvement of our research.

Once again, we would like to express our sincere gratitude for your dedication to reviewing our manuscript and for your invaluable feedback. Your support has been instrumental in shaping the final version of our paper. Thank you for your continued encouragement and guidance.

Best regards,

the authors